# Towards A Generative Protein Evolution Machine with DPLM-Evo

**Xinyou Wang** $^{\diamond\,\heartsuit\,*}$ **Liang Hong** $^{\triangle\,\heartsuit\,*}$ **Jiasheng Ye** $^{\square\,\heartsuit}$ **Zaixiang Zheng** $^{\heartsuit\,\P}$
**Shujian Huang** $^{\diamond\,\ddagger}$ **Quanquan Gu** $^{\heartsuit\,\ddagger}$

## Abstract

Proteins are shaped by gradual evolution under biophysical and functional constraints. Protein language models learn rich evolutionary constraints from large-scale sequences, and discrete diffusion-based protein language models (*e.g.*, DPLMs) are promising for both understanding and generation. However, existing DPLMs typically rely on masked diffusion that contradicts a simple biological intuition: proteins evolve through accumulated edits, not by emerging from masks. Consequently, these frameworks lack explicit pretraining objectives for substitution and insertion/deletion (indel) operations, limiting both optimization-style post-editing and flexible guided generation. To address these limitations, we present DPLM-Evo, an evolutionary discrete diffusion framework that explicitly predicts substitution, insertion, and deletion operations during denoising. DPLM-Evo decouples an upsampled-length latent alignment space from the variable-length observed sequence space, which makes indel-aware generation tractable. To better align substitutions with real evolution, we further introduce a contextualized evolutionary noising kernel that produces biologically informed, context-dependent mutation patterns. Across tasks, DPLM-Evo improves sequence understanding and achieves state-of-the-art mutation effect prediction performance on ProteinGym in the single-sequence setting. It also enables variable-length simulated evolution, and post-editing/optimization of existing proteins

via explicit edit trajectories. Code will be released as part of the roadmap of DPLM Family: https://bytedance.github.io/dplm.

## 1 Introduction

Today's rapidly growing sequence databases archive the results of protein evolution over millions of years, capturing both conserved patterns and extensive natural variation across families. For protein engineering, the practical goal is often not only to generate "protein-like" sequences, but also to leverage this evolutionary information to (i) predict the functional impact of mutations and (ii) propose variants that preserve the structure while improving or reprogramming function.

Protein language models (PLMs) trained on large protein sequence corpus have become a dominant paradigm for learning such evolutionary regularities (Lin et al., 2022; Hayes et al., 2024; Nijkamp et al., 2022; Wang et al., 2024b;a). By modeling the statistics of natural sequence variation, PLMs enable diverse applications including sequence-only zero-shot mutation effect prediction (Meier et al., 2021) and protein structure prediction (Lin et al., 2022). In many real design workflows, however, the problem is inherently *edit-based*: engineers start from a natural scaffold and iteratively introduce substitutions and indels to modify loops, linkers, termini, or binding interfaces while preserving the overall fold and key functional sites.

Recently, discrete diffusion models have attracted increasing attention as generative foundations for protein sequences (Sohl-Dickstein et al., 2015; Austin et al., 2021; Hoogeboom et al., 2021; Zheng et al., 2023a; Sahoo et al., 2024; Shi et al., 2024; Nie et al., 2025). Discrete diffusion-based PLMs (e.g., DPLM (Wang et al., 2024b)) offer a bidirectional receptive field and can capture long-range dependencies that are important for proteins, where epistatic couplings between distant residues often determine stability and function (Wang et al., 2024b). These models have demonstrated strong performance for both understanding and generation (Wang et al., 2024b;a; Alamdari et al., 2023; Hayes et al., 2024).

Despite this progress, most diffusion-based PLMs adopt an *masked diffusion* framework, where masking serves as the noising kernel and generation reduces to iterative masked-

---

$^{*}$Equal contribution $^{\P}$Project Lead $^{\ddagger}$Corresponding authors $^{\diamond}$School of Computer Science, Nanjing University $^{\triangle}$Department of Computer Science and Engineering, CUHK $^{\square}$School of Computer Science, Fudan University $^{\heartsuit}$ByteDance Seed (This work was done during Xinyou, Liang and Jiasheng's internship at ByteDance Seed). Correspondence to: Shujian Huang <huangsj@nju.edu.cn>, Quanquan Gu <quanquan.gu@bytedance.com>.

*Proceedings of the $43^{rd}$ International Conference on Machine Learning*, Seoul, South Korea. PMLR 306, 2026. Copyright 2026 by the author(s).

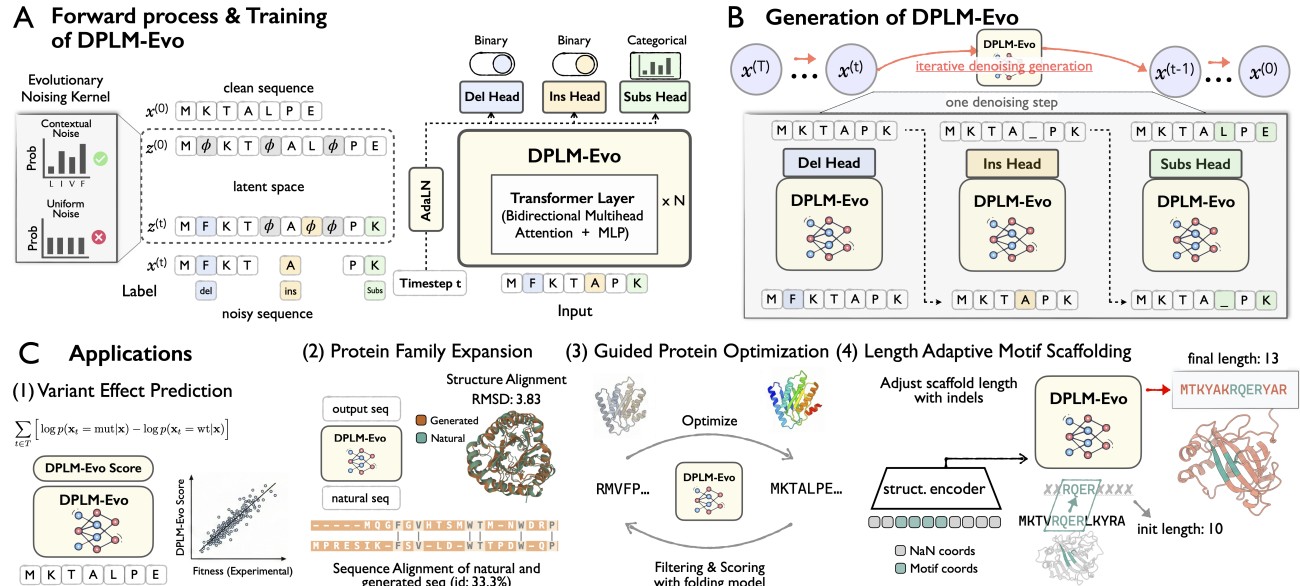

*Figure 1.* (A) DPLM-Evo decouples the upsampled-length latent alignment space from the variable-length observed space. The forward process is performed in latent space. We leverage a random-noising kernel for insertion/deletion and a *contextualized evolutionary noising kernel* for substitution, which generates biologically plausible mutations that are more informative than random noise and encourage the model to capture evolutionary dependencies. DPLM-Evo operates on the noisy observed sequence $x_t$ and employs three heads: a token prediction head for substitution, and binary classification heads for insertion and deletion. (B) Sampling process for DPLM-Evo. This enables unconditional foldable sequence generation via evolutionary denoising (substitution, insertion, and deletion). (C) Applications: 1) State-of-the-art variant effect prediction on the ProteinGym benchmark in the single-sequence setting. 2) Protein family expansion of natural sequences that yields highly distinct sequences while preserving structure. 3) Optimization of existing proteins (e.g., GFP) through iterative directed evolution. 4) Applying DPLM-Evo to conditional generation. In the motif-scaffolding scenario, DPLM-Evo leverages an additional structure encoder to process the coordinates of motif residues. Moreover, DPLM-Evo is capable of adjusting the scaffold length using the insertion and deletion heads during sampling.

token recovery. This is only a loose proxy for biological sequence evolution. Natural evolution proceeds through discrete actions—*substitutions, insertions, and deletions*—that jointly modify both sequence identity and length. Indels are central to remodeling flexible regions (e.g., loops), tuning linker lengths, and creating or removing short motifs that can strongly affect function. A masking-only diffusion process lacks native insertion/deletion actions (Wu et al., 2025; Havasi et al., 2025), and its fixed-length generation makes it awkward to express variable-length evolutionary trajectories or to carry out realistic post-editing of existing proteins. This motivates the following question: *Can we develop a diffusion protein language model that explicitly models evolutionary edit operations (substitution, insertion, and deletion), enabling both more faithful evolutionary modeling and flexible variable-length generation?*

To address this, we propose DPLM-Evo, an evolutionary discrete diffusion framework for protein sequences (Fig. 1). A key challenge is that standard discrete diffusion is defined over a fixed-dimensional categorical state space, making it difficult to model an evolutionary denoising process that modifies sequences through insertions and deletions (*a.k.a.*, indels). Our key idea is to decouple the variable-length observed sequence space from an upsampled-length *latent alignment* space augmented with interleaved gap slots. As

such, the forward diffusion process is defined over the latent alignment space, where indels are represented as gap ↔ residue transitions in the latent sequence. On top of this formulation, DPLM-Evo predicts evolutionary actions explicitly through separate heads for substitution, insertion, and deletion during denoising. Moreover, to make substitution corruption biologically informative, we employ a *contextualized evolutionary noising kernel* that yields data-dependent mutational corruptions conditioned on the surrounding residues, approximating the evolutionary preferences of each site given its sequence context. This leads to biologically informative corruptions rather than uninformative uniformly random substitutions, encouraging the model to capture dependencies consistent with observed evolutionary patterns.

DPLM-Evo unlocks three capabilities that are difficult for prior diffusion PLMs to achieve simultaneously. First, by aligning denoising with explicit evolutionary edits, it improves sequence understanding and provides strong support for mutation effect prediction in a sequence-only setting. Second, it enables flexible and variable-length generation via evolutionary actions, including indels, thereby removing the fixed-length restriction of masking-based diffusion. Third, it supports evolutionary-style post-editing and directed-evolution-style optimization of existing proteins by

generating explicit edit trajectories rather than only filling masked positions.

In summary, our contributions are:

- *Evolutionary discrete diffusion.* We extend discrete diffusion to explicitly model substitution, insertion, and deletion within a unified framework.
- *Length-adaptive modeling with latent alignment.* We decouple an upsampled-length latent alignment space from the variable-length observed sequence space, enabling indel-aware diffusion while retaining efficient computation.
- *Biologically-informed noising for substitution.* We introduce a contextualized evolutionary noising kernel that generates plausible mutational corruptions to improve learning efficiency and consistency with observed evolutionary patterns.
- *Empirical validation across understanding and generation.* We show that DPLM-Evo achieves state-of-the-art mutation effect prediction on ProteinGym in the single-sequence setting, and enables variable-length evolutionary generation and post-editing/optimization of proteins.

## 2 Preliminaries

We briefly review diffusion protein language models (DPLMs) under the masked diffusion framework, which underpins our evolutionary extension.

**DPLM with masked diffusion.** Diffusion protein language model (DPLM, Wang et al., 2024b), in particular, shows excellent performance in both generation and representation learning of protein sequences, and even structures thanks to its recent multimodal extension DPLM-2 (Wang et al., 2024a). The family of DPLMs is grounded in the *masked* discrete diffusion framework (Austin et al., 2021; Zheng et al., 2023a; Sahoo et al., 2024; Liu et al., 2025; Nie et al., 2025), which is characterized by a forward and backward Markov process. Let $\text{Cat}(\mathbf{x}; \mathbf{p})$ be a categorical distribution on protein sequence $\mathbf{x}$ parameterized by a probability vector $\mathbf{p}$ over the vocabulary $\mathcal{A}$ of $K$ amino acids. The forward process of masked diffusion is governed by

$$q(\mathbf{x}_t|\mathbf{x}_0) = \text{Cat}\big(\mathbf{x}_t; \bar{\alpha}_t \delta_{\mathbf{x}_0} + (1 - \bar{\alpha}_t)\pi_{\text{mask}}\big),$$

which gradually perturb the data $\mathbf{x}_0 \sim q(\mathbf{x})$ into an absorbing state $\mathbf{x}_T \sim \pi_{\text{mask}}$. The learned *backward* process $p_\theta(\mathbf{x}_{t-1}|\mathbf{x}_t)$ reversely denoises the $\mathbf{x}_T$ towards the data distribution $\mathbf{x}$, which is typically optimized by the variational bound of the log-likelihood (Ho et al., 2020).

The learning objective of masked diffusion can be simplified into weighted cross-entropies (Zheng et al., 2023a; Sahoo et al., 2024; Shi et al., 2024; Ou et al., 2024), resembling masked language modeling at arbitrary noise levels:

$$\mathcal{L}_t = \mathbb{E}_{q(\mathbf{x})}\text{KL}\big[q(\mathbf{x}_{t-1}|\mathbf{x}_t, \mathbf{x})\|p_\theta(\mathbf{x}_{t-1}|\mathbf{x}_t)\big]$$
$$= \mathbb{E}_{q(\mathbf{x})}\Big[-\lambda_t \sum_{1 \leq i \leq L}\mathbb{I}_{x_t^{(i)} \neq x_0^{(i)}} \cdot \log p_\theta(x_0^{(i)}|\mathbf{x}_t)\Big],$$

where $\lambda_t$ is a weighting coefficient induced by the noising schedule. For inference, DPLM is able to generate amino acid sequences by the reverse iterative denoising process in a *mask-predict* manner, which starts from an all-mask sequence and iterates towards a synthesized sequence. At time $t$, it first generates $\tilde{\mathbf{x}}^{(0)}$ from $p_\theta(\cdot|\mathbf{x}_t)$, then samples a less noisy $\mathbf{x}_{t-1}$ by $q(\cdot|\mathbf{x}_t, \mathbf{x}_0 = \tilde{\mathbf{x}}^{(0)})$.

**Motivation.** DPLM provides a strong generative and representational backbone, but its masked diffusion formulation operates under a fixed-length sequence constraint and casts generation primarily as iterative mask prediction. This design makes it difficult to (i) represent the elementary evolutionary edits that biologists and engineers apply in practice—*substitutions, insertions, and deletions*—and (ii) support flexible, variable-length trajectories during generation and post-editing. These limitations motivate our method: we decouple an upsampled-length latent alignment space from the variable-length observed sequence space and explicitly parameterize substitution, insertion, and deletion actions in both the forward noising and reverse denoising processes. In the next section, we extend the DPLM framework to explicitly model evolutionary edit operations.

## 3 Methodology

Protein evolution and engineering proceed through discrete edit operations—*substitution*, *insertion*, and *deletion*—which naturally induce variable-length sequence trajectories. Standard discrete diffusion models are typically defined on a fixed-length discrete space and therefore cannot explicitly represent indels or variable-length generation. To this end, we extend the standard masked discrete diffusion to an evolutionary discrete diffusion framework that supports explicit edit actions during denoising.

Our key idea is to decouple a variable-length *observed sequence space* from an upsampled-length *latent alignment space* (Havasi et al., 2025; Graves et al., 2006). We define the forward noising and reverse denoising processes in an upsampled-length latent buffer, while the neural network operates on the collapsed observed sequence. During reverse denoising, DPLM-Evo predicts three quantities at each observed token: (i) a substitution distribution over amino acids, (ii) a deletion probability indicating whether the token should be removed, and (iii) an insertion probability indicating whether a new residue should be inserted to its right (and its identity is generated by the substitution head). This design enables explicit edit trajectories and variable-length sampling while keeping computation efficient.

### 3.1 An Evolutionary Discrete Diffusion Framework

**Accommodating length-adaptive *indel* modeling with latent alignment.** Let $\mathcal{V}$ represent the amino acid vocabulary. To handle variable-length sequences within a fixed-dimensional computation framework, we draw inspiration from the latent-alignment methods (Graves et al., 2006;

[Havasi et al., 2025](#)) to distinguish between two spaces:

- **Observed Space** $\mathcal{X}$**:** The set of original protein sequences $\mathbf{x} \in \mathcal{V}^L$, where $L$ is the sequence length, $\mathcal{V} = \mathcal{A} \cup \{\mathbf{m}\}$ with an additional mask token to serve as placeholder for some $j$-th residues with yet-to-be-determined identities.
- **Latent Alignment Space** $\mathcal{Z}$**:** The set of sequences in an upsampled size $N = 2L$, defined over an extended vocabulary $\mathcal{V}^+ = \mathcal{V} \cup \{\phi\}$ with $\phi$ a gap token, which represents an empty slot.

We define a deterministic *collapse function* $\Gamma^{-1}(\mathbf{z}) : \mathcal{Z} \to \mathcal{X}$ that maps a latent alignment $\mathbf{z}$ to an observed sequence $\mathbf{x}$ by removing all $\phi$ tokens, *i.e.*, $\Gamma^{-1}(\mathbf{z}) = [\mathbf{z}^{(j)} \mid \mathbf{z}^{(j)} \neq \phi]_{j=1}^N$. Conversely, $\Gamma(\mathbf{x})$ denotes the set of all possible latent alignments $\mathbf{z}$ that collapse to $\mathbf{x}$, obtained by inserting exactly $L$ gap tokens $\phi$ into $\mathbf{x}$ at arbitrary positions, for example $[A, B, C] \mapsto [A, \phi, \phi, B, \phi, C]$. Under this construction, the latent alignment $\mathbf{z}$ is strictly longer than the observed sequence $\mathbf{x}$, enabling length-changing generation to be carried out on the expanded latent canvas.[1]

Given an observed protein $\mathbf{x}_0$, we treat its alignment $\mathbf{z}_0$ as a latent variable, the training objective is to maximize the evidence lower bound (ELBO) of the log-likelihood:

$$\log p_\theta(\mathbf{x}_0) = \log \sum_{\mathbf{z}_0} p_\theta(\mathbf{x}_0, \mathbf{z}_0) = \log \sum_{\mathbf{z}_0 \in \Gamma(\mathbf{x}_0)} p_\theta(\mathbf{z}_0)$$

$$\geq \mathbb{E}_{\mathbf{z}_0 \in \Gamma(\mathbf{x}_0)} \left[ \mathbb{E}_{\mathbf{z}_t \sim q_t(\mathbf{z}_t|\mathbf{z}_0)} \left[ \log p_\theta(\mathbf{z}_0|\mathbf{z}_t) \right] \right].$$

**Forward noising process for sequence with holistic edit operations.** Unlike predominant masked diffusion models that use absorbing-state (mask) noise, we introduce a new noising prior $\pi(\mathbf{z}_0)$ that respects all possible sequence edit operations.

$$\mathbf{x}_0 \xrightarrow{\Gamma} \mathbf{z}_0 \xrightarrow{\bar{\alpha}_t} \mathbf{z}_t \xrightarrow{\Gamma^{-1}} \mathbf{x}_t$$
$$\mathbf{Q}_{\text{noise}} \downarrow \qquad \nearrow 1-\alpha_t$$
$$\pi(\mathbf{z}_0)$$

Specifically, the forward transition $q(\mathbf{z}_t|\mathbf{z}_0)$ is defined as an interpolant of the original data and the noise distribution:

$$q_t(\mathbf{z}_t|\mathbf{z}_0) = \bar{\alpha}_t \delta_{\mathbf{z}_0} + (1 - \bar{\alpha}_t)\pi(\mathbf{z}_0),$$

where $\bar{\alpha}_t$ is the noise schedule. The noise distribution $\pi(\mathbf{z}_0)$ depends on the initial state $\mathbf{z}_0$ via a transition matrix $\mathbf{Q}_{\text{noise}} \in \mathbb{R}^{(K+2)\times(K+2)}$:

$$\pi(\mathbf{z}_0) = \text{Cat}(\cdot \mid \mathbf{Q}_{\text{noise}}\mathbf{z}_0).$$

To explicitly control the ratio of substitutions, insertions, and deletions, we parameterize $\mathbf{Q}_{\text{noise}}$ with a deletion ratio

---

[1]The $2L$ buffer implies that net insertions cannot exceed $L$. This is sufficient for typical protein engineering workflows but precludes extreme length expansion beyond double the original length.

$\omega_{\text{del}}$ and an insertion ratio $\omega_{\text{ins}}$:

$$\mathbf{Q}_{\text{noise}} = \begin{pmatrix} \overbrace{(1-\omega_{\text{del}})(1-\rho_{\text{mask}})\mathcal{T}_{\text{sub}}}^{z\in\mathcal{A}} & \overbrace{\mathbf{0}_K}^{z=\mathbf{m}} & \overbrace{\omega_{\text{ins}}\frac{1}{K}\mathbf{1}_K}^{z=\phi} \\ (1-\omega_{\text{del}})\rho_{\text{mask}}\mathbf{1}_K^\top & 1 & 0 \\ \omega_{\text{del}}\mathbf{1}_K^\top & 0 & 1-\omega_{\text{ins}} \end{pmatrix},$$

where $\mathbf{1}_K$ denotes a $K$-dim column vector of all ones, and $\mathcal{T}_{\text{sub}}$ defines a substitution matrix of size $K \times K$ over amino acids. Intuitively, this implies:

- If $z_0^{(j)} \in \mathcal{A}$ (amino acid): It is either substituted (with probability $1 - \omega_{\text{del}}$) by another amino acid or mask token (up to $\rho_{\text{mask}}$), or deleted (becomes $\phi$, with probability $\omega_{\text{del}}$).
- If $z_0^{(j)} = \phi$ (gap): It either becomes an inserted amino acid, with probability $\omega_{\text{ins}}$ or remains a gap (with probability $1 - \omega_{\text{ins}}$).

The substitution matrix $\mathcal{T}_{\text{sub}}$ admits several instantiations: (i) $\mathcal{T}_{\text{sub}} = \mathbf{U}_K = \frac{1}{K}\mathbf{1}_K\mathbf{1}_K^\top$ (uniform) recovers the standard uniform kernel; (ii) $\mathcal{T}_{\text{sub}} = \mathbf{M}_{\text{BLOSUM}}$ (§B.1.6) gives a static, biophysically grounded kernel; (iii) $\mathcal{T}_{\text{sub}}^{(j)} = \mathbb{E}_{q'_t(\mathbf{z}'_t|\mathbf{z}_0)}[p_\theta(\cdot|\mathbf{z}'^{\backslash j}_t, \mathbf{m})]$, where $q'^{(j)}_t(\cdot|\mathbf{z}_0) = \bar{\alpha}_t \delta_{\mathbf{z}_0^{(j)}} + (1-\bar{\alpha}_t)\delta_{\mathbf{m}}$ is a masked-diffusion auxiliary process, yields the proposed **data-dependent contextualized kernel**. Preliminary experiments show that the data-dependent contextualized kernel most effectively models evolutionary patterns among the compared kernels (see §B.1.6 for details). Therefore, we adopt option (iii) as our default setting, as described in the following paragraph.

**Simulating biological sequence mutations as evolutionary noising via contextualized on-policy substitution.**

- **Contextualized substitution kernel.** Standard multinomial diffusion models typically employ uninformative uniform noise for corruption, which ignores the biophysical constraints of the protein fitness landscape and leads to inefficient training. To address this, we propose a *contextualized evolutionary noising kernel* that instantiates the substitution noise sampled from a contextualized distribution. Concretely, for each target position $j$ to be corrupted, (1) we sample an auxiliary partially masked context $\mathbf{z}'_t \sim q'_t(\cdot|\mathbf{z}_0)$, then (2) force position $j$ to the mask token and query the model for $p_\theta(\cdot|\mathbf{z}'^{\backslash j}_t, \mathbf{m})$, resulting in $\mathcal{T}_{\text{sub}}^{(j)}$.
- **Warmup stage.** Since the model's prediction is unreliable at the beginning of training, we first warm up the model with a simple data-independent noising kernel (e.g., masking). After this warmup stage, we use the model's own predictions to construct the contextualized noising kernel, yielding biologically plausible and evolutionarily reasonable mutation noise that is more informative than random corruption. Meanwhile, it encourages the model to capture evolutionary and homologous dependencies between amino acids and sequences.

- **Prior distribution.** We further show the prior distribution at $t=T$ (fully noised, $\bar{\alpha}_T=0$). The auxiliary process collapses to a point mass at the all-mask sequence, $q'_T(\cdot|\mathbf{z}_0) = \delta_{\mathbf{m}^L}$. Therefore, for each position $j$, the contextualized kernel reduces to $\mathcal{T}_{\text{sub}}^{(j)} = p_\theta(\cdot|\mathbf{m}^L)$, yielding a learnable residue prior that reflects natural amino-acid statistics rather than a uniform distribution (see §B.1 for the full derivation).

**Connections to existing discrete diffusion.** Our model can recover existing discrete diffusion models by manipulating coefficients in $\mathbf{Q}_{\text{noise}}$. For example, when $\omega_{\text{del}} = 0$ and $\omega_{\text{ins}} = 0$, we disable indels and our model becomes a fixed-length sequence diffusion model. In such circumstances,

- if $\rho_{\text{mask}} = 1$, $\mathbf{Q}_{\text{noise}}$ will always transits any token into $\mathbf{m}$ and our model reduces to classical masked diffusion (Sahoo et al., 2024; Shi et al., 2024).
- if $\rho_{\text{mask}} = 0$, $\mathbf{Q}_{\text{noise}}$ will transits each token into another random token, and our model reduces to classical uniform diffusion (Austin et al., 2021; Schiff et al., 2025).
- if $\rho_{\text{mask}} \in (0, 1)$, $\mathbf{Q}_{\text{noise}}$ will transits each token into either a random token or $\mathbf{m}$, and our model reduces to a generalized diffusion with mixed masked-uniform noising paths (Austin et al., 2021; von Rütte et al., 2025).

With these connections, we can also initialize our model from pretrained masked diffusion-based models, efficiently reprogramming classical discrete diffusion to enable the full spectrum of sequence edit operations. We note that there are also recent work on variable-length diffusion/flow models (e.g., EditFlow (Havasi et al., 2025) and DreamOn (Wu et al., 2025)) for text generation, and Baron et al. (2025) learning to shrink protein sequences.

### 3.2 Training

*Overall Objective.* The final loss function is the expectation over clean data $\mathbf{x}_0$, and the latent alignment sequence $\mathbf{z}_0$ and $\mathbf{z}_t$ and hyperparameters $\gamma$ that balance evolutionary edits:

$$\mathcal{L}_t = \mathbb{E}_{\mathbf{x}_0,\mathbf{z}_0,\mathbf{z}_t}\left[ \sum_{k=1}^{|\Gamma^{-1}(\mathbf{z}_t)|} \lambda_{t-1}\left(\gamma_{\text{sub}}\mathcal{L}_{\text{sub}}^{(k)} + \gamma_{\text{del}}\mathcal{L}_{\text{del}}^{(k)} + \gamma_{\text{ins}}\mathcal{L}_{\text{ins}}^{(k)}\right) \right].$$

The introduced decomposed losses enables precise control over the model's propensity for different evolutionary operations, addressing the inherent class imbalance between frequent substitutions and rare indels in biological evolution.

**The decomposed training objectives.** To make the training tractable, we should solve a critical issue: the diffusion is defined on the latent sequence $\mathbf{z}_t$, but in practice the neural network $f_\theta$ operates on the original sequence $\mathbf{x}_t = \Gamma^{-1}(\mathbf{z}_t)$, which is collapsed by $\mathbf{z}_t$. To bridge this gap, we define the *Index Mapping Function* $\mathcal{I} : \{1,\ldots,L_t\} \to \{1,\ldots,N\}$ such that $\mathcal{I}(k)$ is the index of the $k$-th non-gap token in the latent sequence $\mathbf{z}_t$. Then, we decompose the

loss defined on the latent sequence $\mathbf{z}$ into three mutually exclusive components defined in the observed space $\mathbf{x}$, i.e., substitution loss, deletion loss and insertion loss, based on the token category between $\mathbf{z}_t$ (current noisy state) and $\mathbf{z}_0$ (ground truth).

To more clearly decouple the prediction of the three operations, we leverage separate and independent heads, $p_\theta^{\text{sub}}$, $p_\theta^{\text{del}}$ and $p_\theta^{\text{ins}}$ for the substitution, deletion and insertion prediction for each token in the original sequence $\mathbf{x}_t$. We define the loss for the $k$-th token of the input sequence $\mathbf{x}_t$:

*(1). Substitution Loss.* It is active only when the input and target token are both valid amino acids:

$$\mathcal{L}_{\text{sub}}^{(k)} = \mathbb{I}_{(\mathbf{z}_0^{(\mathcal{I}(k))}\in\mathcal{V})} \cdot \mathbb{I}_{(\mathbf{z}_t^{(\mathcal{I}(k))}\in\mathcal{V})} \cdot \mathbb{I}_{(\mathbf{z}_0^{(\mathcal{I}(k))}\neq\mathbf{z}_t^{(\mathcal{I}(k))})}$$
$$\cdot \text{CE}\left(\mathbf{z}_0^{(\mathcal{I}(k))}, p_\theta^{\text{sub}}(\cdot|\mathbf{x}_t))\right).$$

*(2). Deletion Loss.* It encourages the model to predict $\phi$ if the current token is noise when its target is a gap in $\mathbf{z}_0$:

$$\bar{\mathcal{L}}_{\text{del}}^{(k)} = \mathbb{I}_{(\mathbf{z}_0^{(\mathcal{I}(k))}=\phi)} \cdot \mathbb{I}_{(\mathbf{z}_t^{(\mathcal{I}(k))}\in\mathcal{V})} \cdot \text{CE}\left(\mathbf{z}_0^{(\mathcal{I}(k))}, p_\theta^{\text{del}}(\cdot|\mathbf{x}_t)\right).$$

*(3). Insertion Loss.* Let $v_{\text{next}}^{(k)}$ be the first non-gap token in $\mathbf{z}_0$ between indices $\mathcal{I}(k)$ and $\mathcal{I}(k+1)$. If no such token exists, i.e., there is no insertion needed between $\mathbf{x}_t^k$ and $\mathbf{x}_t^{k+1}$, the $v_{\text{next}}^{(k)}$ is $\emptyset$. The loss is calculated on the positions that need insertion for reconstruction:

$$\bar{\mathcal{L}}_{\text{ins}}^{(k)} = \mathbb{I}_{(v_{\text{next}}^{(k)}\neq\emptyset)} \cdot \text{CE}\left(v_{\text{next}}^{(k)}, p_\theta^{\text{ins}}(\cdot|\mathbf{x}_t)\right).$$

**Practical considerations for $\mathcal{L}_{\text{del}}$ and $\mathcal{L}_{\text{ins}}$.** In our preliminary experiments, we find that training with the original $\bar{\mathcal{L}}_{\text{del}}$ imposes a significant risk of mode collapse, while $\bar{\mathcal{L}}_{\text{ins}}$ leads to unstable training. We manage to solve these issues by instead learning indels as binary classification problems, please refer to Appendix B.2 for more details:

$$\mathcal{L}_{\text{del}}^{(k)} = \text{BCE}(\mathbb{I}_{(\mathbf{z}_0^{(\mathcal{I}(k))}=\phi)}, p_\theta^{\text{del}}(\cdot|\mathbf{x}_t)),$$
$$\mathcal{L}_{\text{ins}}^{(k)} = \text{BCE}(\mathbb{I}_{(v_{\text{next}}^{(k)}\neq\emptyset)}, p_\theta^{\text{ins}}(\cdot|\mathbf{x}_t)).$$

### 3.3 Generation of DPLM-Evo

DPLM-Evo samples with the standard iterative denoising paradigm of discrete diffusion models, while explicitly supporting insertions and deletions. We initialize a fully noisy sequence $\mathbf{x}_T$ by sampling from a learned prior $p_\theta(\cdot|\mathbf{m}^{L_{\text{init}}})$ (see Appendix B.1.5 for details), and iteratively sampling from the following reverse process:

$$p_\theta(\mathbf{x}_{t-1}|\mathbf{x}_t) = \sum_{\mathbf{z}_t\in\Gamma(\mathbf{x}_t)} p(\mathbf{z}_t|\mathbf{x}_t) \sum_{\hat{\mathbf{z}}_0}\sum_{\mathbf{z}_{t-1}\in\Gamma(\mathbf{x}_{t-1})}$$
$$q(\mathbf{z}_{t-1}|\hat{\mathbf{z}}_0,\mathbf{z}_t)p_\theta(\hat{\mathbf{z}}_0|\mathbf{z}_t),$$

where $\mathbf{x}_{t-1} = \Gamma^{-1}(\mathbf{z}_{t-1})$ is obtained deterministically by removing all gap tokens $\phi$ from $\mathbf{z}_{t-1}$. Therefore, the reverse

transition extends from the observed space to the latent alignment space. The transition of the gap token $\phi$ in the latent sequence ($\phi \rightarrow$ AA and AA $\rightarrow \phi$ defined in $\mathbf{Q}_{\text{noise}}$) naturally supports insertions and deletions during sampling. However, exact sampling would require marginalizing over all latent alignments in $\Gamma(\mathbf{x}_t)$, which is intractable. We therefore use a practical approximate sampler by fixing $p(\mathbf{z}_t|\mathbf{x}_t)$ to a canonical alignment with one insertion slot per residue (e.g., $[A, B, C] \mapsto [A, \phi, B, \phi, C, \phi]$), making it a point mass and avoiding enumeration over $\Gamma(\mathbf{x}_t)$. Multi-residue insertions are handled by composing single-slot insertions across successive denoising steps.

We then describe the overall sampling process. We leverage an analogous route-and-denoise factorization (Zheng et al., 2023a) as standard discrete diffusion in the latent space. We maintain a noisy index set $\mathcal{N}_t$ that tracks tokens to be updated at timestep $t$, and predict insertion/deletion operation for each token using the binary heads introduced in §3.2. For each denoising step: (i) **delete** tokens with $p_\theta^{\text{del}}(\mathbf{x}_t^{(j)}) > \tau_{\text{del}}$ for all noisy indices; (ii) **insert** a mask token $\mathbf{m}$ to the right of positions with $p_\theta^{\text{ins}}(\mathbf{x}_t^{(j)}) > \tau_{\text{ins}}$ for all noisy indices; (iii) **substitute** all noisy indices and predict mask tokens introduced by insertion using the substitution head, then define new noisy indices as the least-confident tokens; (iv) **renoise** by sampling from a biologically grounded noising kernel $\pi_{\text{noise}}(\cdot|\mathbf{x}_t)$. Please refer to Appendix C for details.

## 4 Experiments

In this section, we evaluate DPLM-Evo across various understanding and generative tasks. First, we assess variant effect prediction to validate the model's understanding of protein evolution. Subsequently, we examine the model's generation capabilities, including unconditional generation (covering both substitution-only and full edit operations) and conditional motif scaffolding scenario. Finally, we demonstrate the potential application of DPLM-Evo in protein sequence optimization.

### 4.1 Variant Effect Prediction

**Setup.** Modeling the effect of sequence variation on function is fundamental for understanding and designing proteins. DPLM-Evo predicts variant effects using protein sequence only, without supervision from experimental data. **Unlike** the common masked-residue scoring pipeline used by masked language models (i.e., masking the residue(s) of interest and reading out the logits at the masked positions), DPLM-Evo is a substitution-based model that natively scores variants **without masking.** Instead, we directly input the wild-type sequence and evaluate the model's substitution distribution at the mutated site(s), which better matches the model design and avoids introducing an artificial mask token.

For a variant with mutation set $\mathcal{M}$, similar to ESM-1v (Meier et al., 2021), we use a log-odds mutation score

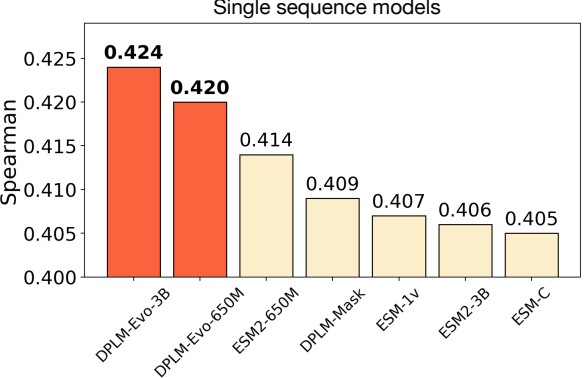

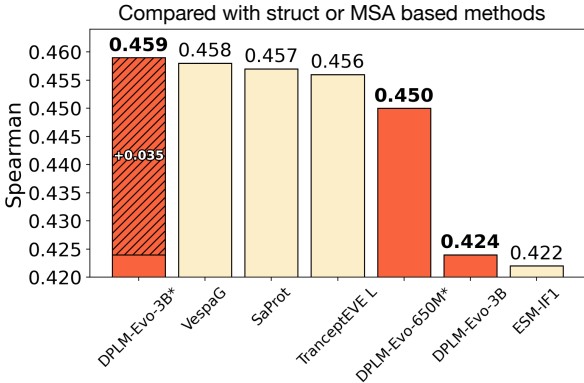

*Figure 2.* ProteinGym variant effect prediction. The * denotes DPLM-Evo after explicitly aligning with the evolutionary kernel.

that compares the mutant residue to the wild-type residue:

$$\sum_{i \in \mathcal{M}} \left[ \log p(\mathbf{x}_i = \text{mut} \mid \mathbf{x}) - \log p(\mathbf{x}_i = \text{wt} \mid \mathbf{x}) \right],$$

where $\mathbf{x}$ denotes the wild-type sequence and $p(\mathbf{x}_i = \cdot \mid \mathbf{x})$ is the substitution probability at position $i$ predicted by DPLM-Evo conditioned on the unmodified wild-type context. In contrast, masked-residue approaches typically score mutations via

$$\sum_{i \in \mathcal{M}} \left[ \log p(\mathbf{x}_i = \text{mut} \mid \mathbf{x}_{\backslash \mathcal{M}}) - \log p(\mathbf{x}_i = \text{wt} \mid \mathbf{x}_{\backslash \mathcal{M}}) \right],$$

where $\mathbf{x}_{\backslash \mathcal{M}}$ indicates that the mutated positions are masked/removed from the input; DPLM-Evo does not require this step.

For the application of DPLM-Evo to the ProteinGym indel benchmark, we first compute the Levenshtein operations (insertions, deletions, substitutions) between wild-type and mutant sequences. The indel score is computed as $\log p(\text{del}) - \log p(\text{keep}) = l$, where $l$ is the deletion logits. This follows from $\log \text{sigmoid}(l) - \log \text{sigmoid}(-l) = l$. An analogous formulation is used for insertions.

**Results.** We evaluate the performance on the ProteinGym DMS substitution zero-shot benchmark (Notin et al., 2023) by calculating the correlation between DPLM-Evo's score and experimental fitness score across all 217 DMS assays. As shown in Fig. 2 (top), *(1) DPLM-Evo achieves the highest correlation score among all the single sequence foun-*

*Table 1.* *ProteinGym indel benchmark results.* DPLM-Evo achieves state-of-the-art among single-sequence methods.

| Method | Input | Avg. Spearman |
|---|---|---|
| ProFam (ensemble) | MSA | 0.530 |
| PoET (Truong Jr & Bepler, 2024) | MSA | 0.517 |
| **DPLM-Evo** | **Sequence** | **0.495** |
| ProGen2 M (Nijkamp et al., 2022) | Sequence | 0.464 |
| RITA L | Sequence | 0.457 |
| Tranception M | Sequence | 0.453 |
| TranceptEVE M | MSA | 0.424 |

**dation models for variant effect prediction in ProteinGym.**
DPLM-Evo outperforms the ESM model series, including ESM-2 (Lin et al., 2022), ESM-C (ESM Team, 2024), ESM-1v (Meier et al., 2021), and DPLM (Wang et al., 2024b). According to Fig. 2 (bottom), DPLM-Evo even surpasses ESM-IF1 (Hsu et al., 2022), which utilizes additional structure information, despite DPLM-Evo using a single sequence (extended structure/MSA methods in Table 2). Crucially, we observe that scaling up the model leads to further improvements, as evidenced by the 3B model outperforming the 650M model. This scalability stands in contrast to ESM-2, which exhibits a performance regression as model size increases (with ESM-2 3B underperforming ESM-2 650M by approximately 0.01 in correlation). We attribute this strong correlation to the model's evolutionarily-inspired pretraining, which fundamentally enables it to learn mutation preferences from natural proteins, effectively capturing the constraints imposed by natural selection.

*(2) Explicitly aligning with evolutionary kernel further unlocks the potential of DPLM-Evo in mutation effect prediction.* We adopted the strategy proposed by VespaG (Marquet et al., 2024) to explicitly align DPLM-Evo output distribution with GEMME (Laine et al., 2019), a state-of-the-art evolutionary-informed prediction model. Leveraging multiple sequence alignment information (Deorowicz et al., 2016), GEMME analyzes the evolutionary mutation sensitivity of individual sites, thereby providing a substitution distribution at each position. By aligning the substitution kernel of DPLM-Evo with that of GEMME, the scores correlate more closely with natural mutations. Illustrated in Fig. 2 (bottom), this alignment yields further enhancements, outperforming SaProt (Su et al., 2023) that takes additional structure, TranceptEVE L with supplementary MSA, and the original VespaG method (based on ESM2-3B). Ablation confirms the contextualized kernel's contribution: replacing it with uniform corruption drops the average Spearman from 0.42 to 0.295.

*(3) DPLM-Evo achieves state-of-the-art indel variant effect prediction among single-sequence methods.*
We further evaluate on the ProteinGym indel benchmark (Notin et al., 2023), directly testing DPLM-Evo's ability to score insertion and deletion variants. As shown in Table 1, DPLM-Evo achieves 0.495 Spearman, surpassing the strongest single-sequence baseline ProGen2 M (Nijkamp

et al., 2022) (0.464) by a significant margin, and approaching MSA-based methods such as PoET (Truong Jr & Bepler, 2024) (0.517) and ProFam ensemble (0.530). This validates that explicit indel modeling in the diffusion framework transfers to improved indel variant effect prediction.

## 4.2 Unconditional Protein Sequence Generation

**Setup.** We initialize DPLM-Evo from a pretrained DPLM-650M model (Wang et al., 2024b). Specifically, the backbone parameters (token embeddings and Transformer blocks) and the substitution head are initialized from DPLM, while the two binary operation heads for indel prediction (deletion and insertion) are randomly initialized. The substitution head reuses the pretrained DPLM output projection (tied with input embeddings); the deletion and insertion heads are each single-layer binary classifiers with negligible parameter overhead. We train on the UniRef50 dataset for 100,000 steps, using 2,000 warmup steps to a peak learning rate of $10^{-4}$, followed by linear decay to $0.1 \times 10^{-4}$ by the end of training. Training uses 32 H100 GPUs for approximately 25 hours. The contextualized kernel adds +24% per-step overhead due to an additional gradient-free forward pass. The diffusion timestep is set to $T = 500$. For unconditional generation, we consider initial lengths $L_{\text{init}} \in \{100, 200, 300, 400, 500\}$.

**Results.** DPLM-Evo performs iterative denoising by jointly applying substitution, deletion, and insertion, enabling variable-length generation that more closely mirrors natural evolutionary trajectories. DPLM-Evo generation starts from corrupted sequences sampled from the learnable diffusion prior rather than all-mask initialization used in masked diffusion. Fig. 3 demonstrates the evaluation results of unconditional generation in various perspectives: (1) *Diversity and Foldability:* Fig. 3A shows DPLM-Evo achieves consistent high foldability across length as measured by ESMFold pLDDT. Quantitatively, DPLM-Evo achieves 83.6 pLDDT, competitive with DPLM (84.0) and DiMA (Meshchaninov et al., 2024) (83.3), and substantially above prior generative models (Table 3). Secondary structure analysis further confirms that generated sequences match natural helix/sheet/loop proportions of SwissProt (Fig. 7). Compared with DPLM based on masked diffusion, Fig. 3D-E shows DPLM-Evo achieves comparable foldability while possessing greater generation diversity, reflected by a larger number of clusters in both sequence and structure. (2) *Reduced Mode Collapse:* DPLM-Evo produces higher sequence entropy than DPLM, as is shown in Fig. 3F, indicating fewer repetition patterns and alleviating the mode collapse issue. (3) *Effect of Evolutionary Kernel:* Training with the *contextualized evolutionary noising kernel* substantially outperforms uniform noising, as shown in Fig. 3D. This indicates that biologically grounded corruptions encourages DPLM-Evo to learn more evolutionarily plausible substitution predictions, yielding higher-quality samples at gen-

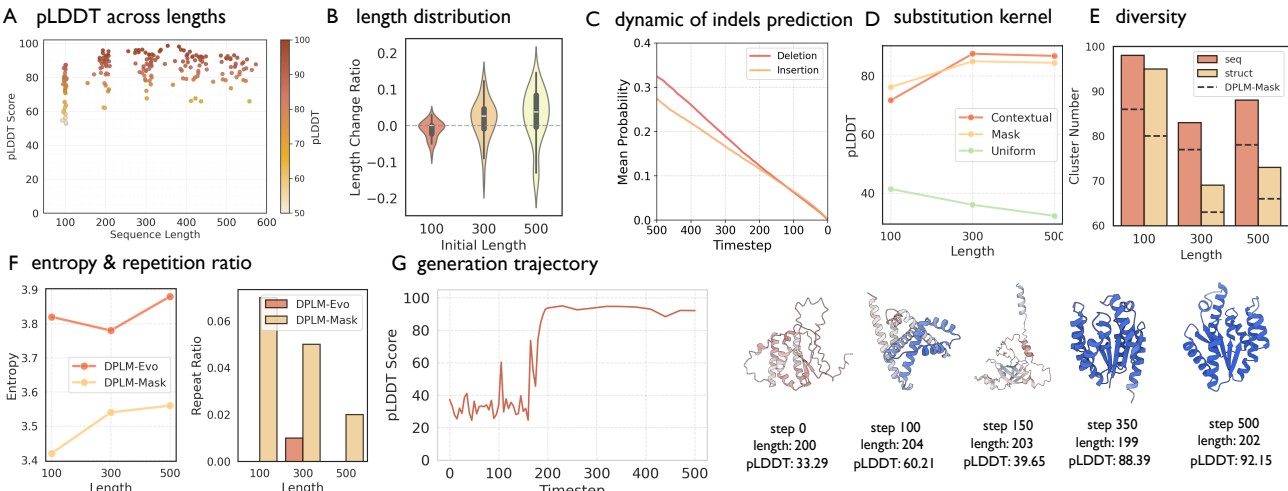

*Figure 3.* Evaluation of unconditional sequence generation by simulating evolution process with DPLM-Evo. (A) Unconditional generation from length 100 to 500 evaluated by pLDDT. (B) Length distribution of DPLM-Evo generations from fixed initial lengths. (C) Insertion/deletion head predicted probability under different timesteps. (D) Ablation on different substitution kernels. We train the same DPLM-Evo model with different substitution kernels and mask kernel significantly outperforms the uniform kernel, but underperforms the contextualized evolutionary kernel. (E) Sequence and structure diversity of DPLM-Evo compared with DPLM-Mask in different lengths. (F) Entropy and repetition comparison between DPLM-Evo and DPLM-Mask. DPLM-Evo outputs sequences with high entropy and close to zero repetition ratio, alleviating the common mode collapse issue. (G) Demonstration of the generation trajectory.

eration time. (4) ***Length Control:*** Output lengths remain concentrated near their initial values without excessive expansion or collapse. The distribution is visualized in Fig. 3B. This indicates that insertion and deletion prediction are invoked conservatively, resulting in a refinement process that prioritizes substitutions over drastic length changes.

To better understand how indel operations are scheduled over the diffusion trajectory, we probe the deletion and insertion heads across timesteps on a representative natural sequence (Fig. 3C). The predicted indel probabilities decrease as the timestep decreases towards clean, suggesting the model primarily uses indels for coarse adjustments during high-noise stages, and gradually shifts to fine-grained refinements later. This behavior indicates that indel operations can be manipulated through timestep control, e.g., fixing the deletion timestep to 0 for insertion-only tasks.

### 4.3 Length-adaptive Scaffolding of Functional Motifs

**Setup.** Motif scaffolding aims to generate a protein scaffold for a given functional motif. We evaluate DPLM-Evo in *zero-shot* and *continued finetuning* settings. For finetuning, DPLM-Evo incorporates structural constraints for motif structure features, as illustrated in Fig. 1C(4). During generation, DPLM-Evo edits only the scaffold region and never modifies motif residues, allowing dynamic length adjustment to better accommodate the motif. In contrast, fixed-length sequence models require manually scaffold length enumeration and cannot revise length once an unsuitable initialization is chosen. For each motif instance, we sample 100 candidate scaffolds. Success is defined as pLDDT > 70, and motif RMSD < 1 Å.

**Results.** As shown in Fig. 4, in the zero-shot setting, DPLM-

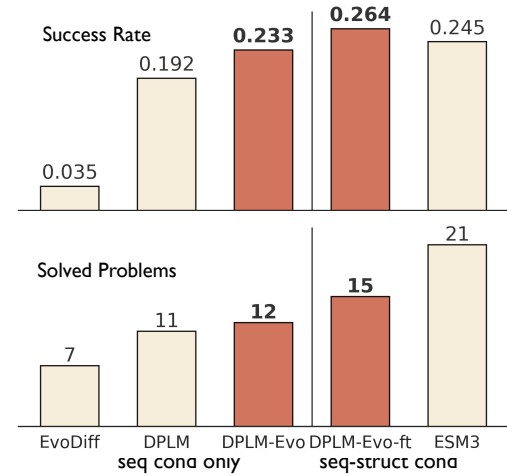

*Figure 4. Evaluation of motif scaffolding.* DPLM-Evo improves both success rates and solved motif counts.

Evo solves more motif problems than EvoDiff and DPLM-Mask, and achieves higher overall success rate (0.23). We attribute this to the capability for dynamic scaffolding length adjustment and evolutionarily plausible mutations provided by the substitution head. Continued finetuning brings further improvements, highlighting the importance of multimodal conditioning. Compared to multi-modal models like ESM-3 (Hayes et al., 2024), the finetuned DPLM-Evo achieves a higher overall success rate but resolves slightly less targets. We hypothesize this gap arises because DPLM-Evo only supports multimodal conditioning, without native end-to-end training for structural understanding. We leave multimodal evolutionary discrete diffusion modeling as an exciting direction for future work.

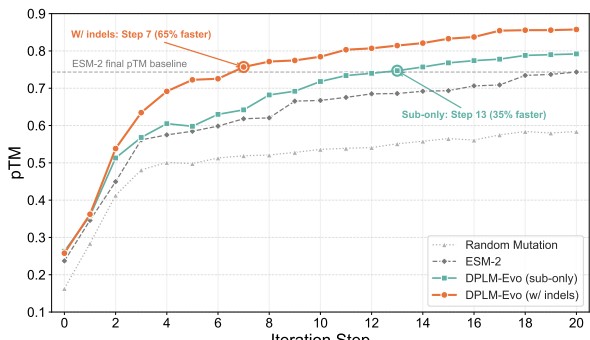

sample 50 candidates starting from the original sequence

| Iter number | plddt | | Clusters | | | |
| --- | --- | --- | --- | --- | --- | --- |
| | mean/std | min/max | Seq Id 0.5 | Seq Id 0.4 | Seq Id 0.3 | Struct TM 0.5 |
| 0 (original) | 87.43 | -/- | - | - | - | - |
| 100 | 83.75/2.8 | 74.75/88.31 | 51 | 37 | 4 | 1 |

RMSD: 3.83          Seq Identity: 33.33%

*Figure 5. Unconstrained in-silico family expansion.* DPLM-Evo preserves the fold despite large sequence edits.

*Figure 6. Directed evolution of GFP. (A) pTM and RMSD through iterations. DPLM-Evo reaches pTM 0.793 (substitution-only) and 0.857 (with indels), compared to ESM-2 at 0.737 and random mutation at 0.6. (B-C) Starting template and optimized structure.*

### 4.4 Case Study: In-silico Sequence Family Expansion

**Setup.** To assess whether DPLM-Evo can generate diverse yet structurally consistent relatives of a given protein, we perform unconstrained post-editing starting from natural sequences. Specifically, we randomly select sequences from the CAMEO dataset and let DPLM-Evo refine them without imposing explicit functional constraints. We evaluate both structural preservation relative to the starting sequence and sequence diversification.

**Results.** DPLM-Evo generates diverse, yet structurally similar protein sequences in the unconstrained optimization setting. **Structural preservation:** We find that DPLM-Evo preserves structural plausibility (evaluated via comparing predicted structure to wild type structure) and at the same time introduces substantial edits. While it does not necessarily increase the initial pLDDT, it effectively explores the sequence space around a given fold without catastrophic structural degradation. **Sequence diversification and family expansion:** Meanwhile, DPLM-Evo modifies a large proportion of the initial sequence (with sequence identity mostly below 50%). Fig. 5 shows a case where the highly modified sequence still aligns structurally with the original. These results suggest that DPLM-Evo performs unconstrained sequence optimization that preserves a shared structural scaffold while producing diverse sequences. This implies that DPLM-Evo captures latent regularities of natural proteins, including constraints related to fold and stability. In this way, the generated sequences can be potentially viewed as in silico expanded homologs of the starting protein, holding the potential for purely sequence based orphan protein understanding.

### 4.5 Case Study: Directed Evolution of GFP

**Setup.** We optimize the green fluorescent protein (GFP) via directed evolution using DPLM-Evo as illustrated in Algorithm 2. Starting from the template, we adopt the beam search strategy to maintain a candidate set: in each iteration, 10 optimized sequences are generated for each sequence in the candidate set. In each step, we employ the Chai-1 model for filtering and structure scoring to keep only the top-scoring candidates retained for the next iteration. Following ESM3 (Hayes et al., 2024), The criteria for filter is that the template chromophore site RMSD is less than 1.5Å, while the scoring term is the pTM score produced by Chai-1.

**Results.** Fig. 6A depicts the trajectory of optimization. We observed that as the iteration processes, the pTM value gradually increases, while the RMSD remained consistently below 1.5Å. After 20 iterations, the pTM increased from an initial 0.263 to 0.793 (substitution-only) while a random mutation baseline converges below 0.6. With indel sampling enabled, pTM further improves to 0.857, demonstrating the benefit of indel-aware optimization. As a comparison, an ESM-2 baseline (Lin et al., 2022) reaches 0.737 under the same beam-search protocol (Fig. 6). Fig. 6B and Fig. 6C visualize the structures of the GFP before and after optimization, respectively. Residues are colored according to their pLDDT value, indicating a significant increase in stability. These results demonstrate that DPLM-Evo can leverage the priors learned from evolution-scale protein sequences to optimize the GFP sequence toward greater overall structural stability, while preserving the structure of the chromophore site to maintain its fluorescence effect.

## 5 Discussion

In this work, we presented DPLM-Evo, a unified framework for evolutionary discrete diffusion that explicitly models substitutions, insertions, and deletions. We decoupled the upsampled-length latent space from variable-length observed sequences, enabling efficient indel-aware diffusion. We further enhanced the learning efficiency and evolutionary consistency of the model through a contextualized noising kernel. Extensive empirical evaluations demonstrate that DPLM-Evo not only achieves state-of-the-art performance in mutation effect prediction on ProteinGym but also opens new avenues for variable-length protein generation and optimization, bridging the gap between deep generative modeling and evolutionary biology.

## Impact Statement

This paper presents work whose goal is to advance the field of machine learning and its applications on protein modeling. Our work on protein generation and representation learning can be used in developing potent therapeutic macromolecules such as antibodies and accelerate the research process of drug discovery. Our method may be adapted to other scenarios of computer-aided design, such as small molecule and material design.

## Acknowledgements

We thank anonymous reviewers for their inspiring feedback. We would like to especially thank Dr. Hang Li for insightful discussions on the project and feedback on the manuscript that help shape this study. We also thank Yuning Shen, Yi Zhou, Wei Qu, Chan Lu as well as other colleagues at ByteDance Seed for their valuable comments and support.

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

# A    Limitations

While our approach achieves promising results, several limitations of the current framework that suggest directions for future work. **(1)** Our canonical alignment restricts each denoising step to at most one insertion or deletion per position, requiring multi-residue indels to be composed across successive steps. Extending the framework to support atomic multi-residue edits could better capture the frequency of large-scale indels in natural evolution. **(2)** On the theoretical side, Algorithm 1 employs practical approximations that fixing the latent alignment to a canonical form and formulating indels as binary classification, rather than exactly marginalizing over all alignments. Developing tighter variational bounds or more expressive alignment distributions remains an open direction. The learnable prior $p_\theta(\cdot|\mathbf{m}^L)$ strictly holds only for the substitution component; under non-zero indel rates with $\omega_{\text{ins}} = \omega_{\text{del}}$, only the expected length $\mathbb{E}[|x_T|] = L$ is preserved rather than the full prior distribution (see details in Appendix B.1.5). **(3)** Finally, the contextualized evolutionary noising kernel introduces about +24% per-step training overhead due to an additional gradient-free forward pass. Exploring more efficient implementations or amortized variants could reduce this cost while retaining the benefits of biologically informed corruption.

# B    Training Details.

## B.1    Substitution Learning with Contextualized Evolutionary Noise

The quality of the DPLM-Evo heavily depends on how the substitution process is modeled. In this section, we discuss the noise kernel for the substitution head.

### B.1.1    MOTIVATION: FROM UNIFORM NOISE TO BIOLOGICAL MANIFOLDS

Standard discrete diffusion models typically employ a uniform noise kernel (Hoogeboom et al., 2021). In the context of protein engineering, this implies that a mutation from a hydrophobic residue (e.g., Leucine) to a charged residue (e.g., Arginine) is as probable as a mutation to another hydrophobic residue (e.g., Isoleucine). This assumption fundamentally contradicts the biophysical constraints of the protein fitness landscape. Training with uniform noise suffers from significant inefficiency: the model spends a large portion of training correcting biologically obvious errors (e.g., restoring a hydrophobic core disrupted by charged noise) rather than learning evolutionary dependencies. To address this, we propose a **contextualized evolutionary noising kernel**. Instead of corrupting data towards random chaos, we utilize the model's own prediction capability to generate noise that is more likely to remain near the natural-protein manifold. This not only provides more informative noisy tokens, which is helpful for denoising, but also encourages the model to capture the dependencies between the wild-type sequence and plausible homolog-like variants generated by model prediction.

### B.1.2    FORMALIZATION: CONSTRUCT THE CONFIDENCE-AWARE KERNEL WITH MASK PREDICTION

We leverage mask token $\mathbf{m}$ (distinct from the gap token $\phi$ used for deletions) to represent unknown semantic identity in the protein sequence. Let $p_\theta$ denote the neural network parameterized by $\theta$. At noise level $t$, the contextualized substitution kernel for an amino acid at index $j$ (where $\mathbf{z}_0^{(j)} \in \mathcal{A}$) follows the main-text definition (option (iii) in §3.1): it is the model's masked prediction at position $j$, averaged over an auxiliary masked-diffusion process $q_t'$ that partially corrupts the surrounding context,

$$\mathcal{T}_{\text{sub}}^{(j)} = \mathbb{E}_{q_t'(\mathbf{z}_t'|\mathbf{z}_0)}\big[\, p_\theta\big(\cdot \mid \mathbf{z}_t'^{\setminus j}, \mathbf{m}\big)\,\big], \tag{1}$$

where $\mathbf{z}_t' \sim q_t'(\cdot|\mathbf{z}_0)$ is a partially masked context with per-position marginal $q_t'^{(j)}(\cdot|\mathbf{z}_0) = \bar{\alpha}_t \delta_{\mathbf{z}_0^{(j)}} + (1 - \bar{\alpha}_t)\delta_{\mathbf{m}}$, and $(\mathbf{z}_t'^{\setminus j}, \mathbf{m})$ denotes this context with position $j$ additionally forced to the mask token. In practice we approximate the expectation with a single sample of $\mathbf{z}_t'$, so that one gradient-free forward pass yields the detached masked predictions $p_\theta(\cdot \mid \mathbf{z}_t'^{\setminus j}, \mathbf{m})$ at all corrupted positions simultaneously. Contextualized noise is then sampled from $\mathcal{T}_{\text{sub}}^{(j)}$ to corrupt the data, and the model is trained to denoise it.

### B.1.3    IMPLEMENTATION: ON-POLICY CONFIDENCE-AWARE KERNEL

We leverage $p_\theta^{\text{sub}}$ of DPLM-Evo for the mask prediction to form $\mathcal{T}_{\text{sub}}^{(j)}$. However, during training, the model is trained to denoise based on the contextualized noise, which consists of the standard amino acids. As training processes, DPLM-Evo will lose its mask prediction capabilities. To prevent catastrophic forgetting of the masking prediction, we construct a mixing noising kernel for the contextualized noise and the mask state. Therefore, DPLM-Evo can learn the mask prediction explicitly during training. Concretely, the noising kernel $\pi$ applies a *confidence-aware* gate that mixes the contextualized substitution kernel $\mathcal{T}_{\text{sub}}^{(j)}$ from Eq. equation 1 with the mask state $\delta_{\mathbf{m}}$ according to the model's confidence:

$$\pi(v \mid \mathbf{z}_0^{(j)}) = \mathbb{I}(c_j > \tau) \cdot \mathcal{T}_{\text{sub}}^{(j)}(v) + \mathbb{I}(c_j \leq \tau) \cdot \delta_{\mathbf{m}}(v) \tag{2}$$

Here, $c_j = \max_{v \in \mathcal{A}} \mathcal{T}_{\text{sub}}^{(j)}(v)$ is the model's confidence in the contextualized substitution at position $j$, and $\tau$ is the confidence threshold for masking. Rather than fixing $\tau$, we treat $\rho_{\text{mask}}$ from $\mathbf{Q}_{\text{noise}}$ (§3.1) as the target mask fraction and *derive* $\tau$ from it: at each step we set $\tau = \text{Quantile}_{\rho_{\text{mask}}}(\{c_j\})$, i.e., the value for which a fraction $\rho_{\text{mask}}$ of the per-position confidences satisfy $c_j \leq \tau$. If $c_j \leq \tau$, this indicates the low confidence prediction that the model is uncertain, which represents insufficiently valuable or evolutionarily relevant information. Therefore, we fallback to the mask token $\mathbf{m}$ for these positions. This reinforces the fundamental masked prediction objective, ensuring the model remains robust to missing information and avoid forgetting about mask prediction.

Crucially, this process is **on-policy**: the noise is generated by the model state $\theta$ with stop-gradient at the current training step. To enhance the quality of contextualized noise at the early training stage and prevent the training instability, we initialize the model parameters from a pre-trained MLM-based pLM or an absorbing discrete diffusion-based pLM.

### B.1.4 BIOLOGICAL INTERPRETATION: TRAVERSING THE FITNESS LANDSCAPE

By replacing the static uniform kernel with our dynamic contextualized kernel, we reframe the diffusion training process as a traversal on the fitness landscape:

1. **Denoising as Error Correction (Lethal Mutations):** When the contextualized kernel $\mathcal{T}_{\text{sub}}^{(j)}$ generates a residue that violates structural constraints (e.g., steric clashes), the training objective is to enable the model to identify and correct these erroneous mutations that are evolutionarily unacceptable.

2. **Denoising as Homology Modeling (Neutral Mutations):** The contextualized noise sampled from $\mathcal{T}_{\text{sub}}^{(j)}$ may also be biologically plausible substitutions. In this regime, the ground truth $\mathbf{x}_0$ represents a specific instance (wild type), while the noise $\mathbf{x}_t$ represents a plausible homolog-like neighboring variant. The denoising loss encourages the model to learn the *evolutionary dependencies* between the original functional sites $\mathbf{x}_0$ and variable sites $\mathbf{x}_t$.

### B.1.5 THE LEARNABLE PRIOR

We discuss the sampling prior of the contextualized noising kernel when $t = T$. In standard multinomial diffusion, the sampling prior is a uniform distribution $\mathcal{U}(\mathcal{V})$, which is a poor approximation of natural proteins. In our framework, the effective prior at $T$ steps becomes the model's prediction given a fully masked sequence:

$$p(\mathbf{z}_T) \approx p_\theta(\cdot \mid \mathbf{m}^L) \tag{3}$$

This **learnable prior** captures the natural background frequencies of amino acids and global sequence statistics (e.g., length distributions and domain compositions) inherent in the pre-trained weights. Consequently, the reverse generation process initializes from a biologically informed distribution rather than random chaos, improving sampling efficiency and stability.

**Formal justification.** Consider the contextualized substitution matrix $\mathcal{T}_{\text{sub}}^{(j)} = \mathbb{E}_{q'_t(\mathbf{z}'_t | \mathbf{z}_0)}[p_\theta(v \mid \mathbf{z}'^{\backslash j}_t, \mathbf{m})]$, where $q'^{(j)}_t(\cdot | \mathbf{z}_0) = \bar{\alpha}_t \delta_{\mathbf{z}_0^{(j)}} + (1 - \bar{\alpha}_t) \delta_{\mathbf{m}}$ is an auxiliary masked-diffusion process. The forward marginal at position $j$ is $q^{(j)}_t(v | \mathbf{z}_0) = \bar{\alpha}_t \delta_{\mathbf{z}_0^{(j)}}(v) + (1 - \bar{\alpha}_t) \mathcal{T}_{\text{sub}}^{(j)}$. At $t = T$ (fully noised, $\bar{\alpha}_T = 0$), we have $q'_T(\cdot | \mathbf{z}_0) = \delta_{\mathbf{m}^L}$ deterministically. Therefore, $q^{(j)}_T(v) = p_\theta(v | \mathbf{m}^L)$ by definition, confirming that the prior under the contextualized kernel equals the model's marginal prediction given a fully masked sequence.

**Prior distribution under non-zero indel rates.** The above argument strictly holds for the substitution component. Under non-zero indel rates ($\omega_{\text{ins}}, \omega_{\text{del}} > 0$), $p_\theta(\cdot | \mathbf{m}^L)$ does not exactly capture the full prior distribution over variable-length sequences. However, when the indel rates are symmetric ($\omega_{\text{ins}} = \omega_{\text{del}}$, as used in our training), the expected observed length is preserved. Starting from a length-$L$ sequence, the canonical latent alignment has $L$ amino acid positions and $L$ gap positions. At step $t$, each amino acid remains non-gap with probability $1 - (1 - \bar{\alpha}_t)\omega_{\text{del}}$, while each gap becomes non-gap with probability $(1 - \bar{\alpha}_t)\omega_{\text{ins}}$. Therefore:

$$\mathbb{E}[|x_t|] = L \cdot \left(1 - (1 - \bar{\alpha}_t)\omega_{\text{del}}\right) + L \cdot (1 - \bar{\alpha}_t)\omega_{\text{ins}} = L \cdot \left[1 + (1 - \bar{\alpha}_t)(\omega_{\text{ins}} - \omega_{\text{del}})\right]. \tag{4}$$

When $\omega_{\text{ins}} = \omega_{\text{del}}$, $\mathbb{E}[|x_t|] = L$ for all $t$, including $t = T$. This only justifies the expected length, not the full prior distribution. In practice, we use $p_\theta(\cdot | \mathbf{m}^{L_{\text{init}}})$ as the generation prior and rely on the insertion/deletion heads during reverse denoising to handle the remaining length variation.

### B.1.6 OTHER ALTERNATIVE NOISING KERNEL: BLOSUM-INFORMED SUBSTITUTION

The contextualized kernel incurs an extra forward pass; for efficiency, we also provide a static, biologically grounded alternative based on the BLOSUM substitution matrices.

Standard discrete diffusion typically uses a uniform transition matrix $\mathbf{U}_K$, which implies that all amino acid substitutions are equally probable. In contrast, the BLOSUM62 matrix encodes empirical substitution frequencies observed in homologous protein alignments. Let $\mathbf{B} \in \mathbb{R}^{K \times K}$ be the BLOSUM62 scoring matrix, where $\mathbf{B}_{ij}$ represents the log-odds score of substituting amino acid $i$ with $j$.

We construct the static substitution noise matrix $\mathbf{M}_{\text{BLOSUM}}$ by applying a row-wise Softmax operation over the scaled scores:

$$[\mathbf{M}_{\text{BLOSUM}}]_{ij} = \frac{\exp(\mathbf{B}_{ij}/\tau)}{\sum_{k=1}^{K} \exp(\mathbf{B}_{ik}/\tau)} \tag{5}$$

where $\tau > 0$ is a temperature hyperparameter that controls the entropy of the noise distribution:

- As $\tau \to \infty$, the distribution approaches the uniform distribution $\mathbf{U}_K$.
- As $\tau \to 0$, the distribution collapses to the identity matrix (no mutation).
- At moderate $\tau$, the distribution favors physico-chemically conservative mutations (e.g., $L \leftrightarrow I$) over radical changes (e.g., $L \leftrightarrow K$).

This static kernel plugs directly into our framework by replacing the uniform component of $\mathbf{Q}_{\text{noise}}$; though less expressive than the contextualized kernel, it still improves over uniform noise by respecting biochemical properties.

**Empirical study.** We compare the uniform, BLOSUM, and contextualized kernels on ProteinGym (average Spearman) using the same dataset and training hyperparameters. The result is: uniform $0.295 <$ BLOSUM $0.35 <$ contextualized $0.42$. Uniform noise carries no evolutionary signal and yields biologically implausible noisy inputs from which the model can hardly learn meaningful evolutionary relationships. BLOSUM provides statistically grounded mutation probabilities but is data-independent (depends only on the current residue), so it lacks global and sequence-level context. The contextualized kernel instead produces data-dependent noise specific to each sequence and evolutionarily meaningful. This allows the model to learn how the same amino acid mutates differently in different protein contexts, effectively capturing context-dependent evolutionary patterns, and helps it ultimately achieve the best performance on the mutational effect prediction task.

### B.2 Deletion and insertion training with binary classification head

We find that only training $\mathcal{L}_{\text{del}}$ to predict the $\phi$ token poses a significant risk of mode collapse. Without exposure to tokens that should *not* be deleted, the head may converge to a degenerate solution that always predicts the gap token $\phi$ for any input, resulting in excessive deletion during generation.

Therefore, we introduce negative samples that represents tokens that should be preserved for deletion training. Given that there are only two distinct prediction targets: either a gap token or retaining the original token, deletion is essentially a *binary classification* task. Therefore, we parameterize the deletion head as a binary prediction head, and define the binary deletion target $y_k^{\text{del}} = \mathbb{I}(\mathbf{z}_0^{(\mathcal{I}(k))} = \phi)$. Similarly, we parameterize the insertion head as a binary classifier. For each token position $k$ in the observed sequence $\mathbf{x}_t$, the head predicts whether an additional token should be inserted immediately to the right of $\mathbf{x}_t^{(k)}$. We define the binary insertion target as $y_k^{\text{ins}} = \mathbb{I}(v_{\text{next}}^{(k)} \neq \emptyset)$, and train both heads with BCE objectives:

$$\mathcal{L}_{\text{del}} = \sum_{k=1}^{|\Gamma^{-1}(\mathbf{z}_t)|} \text{BCE}\left(y_k^{\text{del}}, p_\theta^{\text{del}}(\cdot|\mathbf{x}_t)\right), \tag{6}$$

$$\mathcal{L}_{\text{ins}} = \sum_{k=1}^{|\Gamma^{-1}(\mathbf{z}_t)|} \text{BCE}\left(y_k^{\text{ins}}, p_\theta^{\text{ins}}(\cdot|\mathbf{x}_t)\right). \tag{7}$$

where $\text{BCE}(y, p) = -[y \log p + (1-y) \log(1-p)]$. The insertion head only predicts whether an insertion is needed. When an insertion is triggered during sampling, we first insert a special mask token $\mathbf{m}$ as a noisy placeholder at this location, and then reuse the substitution head for filling the masked position.

## C  Sampling Details

### C.1  Overview

We initialize a fully noisy sequence $\mathbf{x}_T$ by sampling from a learned prior $p_\theta(\cdot|\mathbf{m}^{L_{\text{init}}})$ (see Appendix B.1.5 for details), and iteratively sampling from the following reverse process:

$$p_\theta(\mathbf{x}_{t-1}|\mathbf{x}_t) = \sum_{\mathbf{z}_t, \mathbf{z}_{t-1}, \hat{\mathbf{z}}_0} p_\theta(\mathbf{x}_{t-1}, \mathbf{z}_t, \mathbf{z}_{t-1}, \hat{\mathbf{z}}_0|\mathbf{x}_t)$$

$$= \sum_{\mathbf{z}_t, \mathbf{z}_{t-1}, \hat{\mathbf{z}}_0} p(\mathbf{x}_{t-1}|\mathbf{z}_{t-1}, \mathbf{x}_t) q(\mathbf{z}_{t-1}|\hat{\mathbf{z}}_0, \mathbf{z}_t, \mathbf{x}_t) p_\theta(\hat{\mathbf{z}}_0|\mathbf{z}_t, \mathbf{x}_t) p(\mathbf{z}_t|\mathbf{x}_t)$$

$$= \sum_{\mathbf{z}_t \in \Gamma(\mathbf{x}_t)} p(\mathbf{z}_t|\mathbf{x}_t) \sum_{\hat{\mathbf{z}}_0} \sum_{\mathbf{z}_{t-1} \in \Gamma(\mathbf{x}_{t-1})} q(\mathbf{z}_{t-1}|\hat{\mathbf{z}}_0, \mathbf{z}_t) p_\theta(\hat{\mathbf{z}}_0|\mathbf{z}_t),$$

where $\mathbf{x}_{t-1} = \Gamma^{-1}(\mathbf{z}_{t-1})$ is obtained deterministically by removing all gap tokens $\phi$ from $\mathbf{z}_{t-1}$.

**Algorithm 1 as an approximate sampler.** To make sampling tractable, we make three simplifying choices:

1. *Canonical alignment*: We fix $p(\mathbf{z}_t|\mathbf{x}_t)$ to a deterministic canonical form with one insertion slot per residue (e.g., $[\mathrm{A}, \mathrm{B}, \mathrm{C}] \mapsto [\mathrm{A}, \phi, \mathrm{B}, \phi, \mathrm{C}, \phi]$), eliminating the sum over $\Gamma(\mathbf{x}_t)$.

2. *Single-step indels*: Each denoising step allows at most one insertion per position; multi-residue insertions are composed across successive reverse steps (empirically supported by Fig. 3B–C).

3. *Binary classification*: Indel decisions use binary thresholds $\tau_{\mathrm{ins}} = \tau_{\mathrm{del}} = 0.7$ (practical inference hyperparameters that favor conservative edit decisions) to discretize predicted edit probabilities, avoiding mode collapse (§B.2).

The generation process of DPLM-Evo follows the standard iterative denoising paradigm of discrete diffusion models. Starting from a prior length $L_{\mathrm{init}}$, we initialize $\mathbf{x}_T$ from the learned prior $p_\theta(\cdot|\mathbf{m}^{L_{\mathrm{init}}})$ and mark all positions as noisy. At each reverse step, we first lift the observed sequence $\mathbf{x}_t$ to a canonical latent alignment $\mathbf{z}_t = \Gamma_{\mathrm{can}}(\mathbf{x}_t)$ with one gap slot after each residue, perform the approximate reverse update in the latent space, and then collapse the updated latent sequence back to the observed space:

$$\mathbf{z}_t = \Gamma_{\mathrm{can}}(\mathbf{x}_t), \qquad \mathbf{z}_{t-1} \sim \tilde{p}_\theta(\cdot \mid \mathbf{z}_t, \mathbf{x}_t), \qquad \mathbf{x}_{t-1} = \Gamma^{-1}(\mathbf{z}_{t-1}). \tag{8}$$

Here $\tilde{p}_\theta$ denotes the practical approximate reverse kernel induced by the three prediction heads. Although the reverse kernel is defined in the latent alignment space, we can implement it directly in the observed sequence space under the canonical-alignment approximation. This is because each observed token is paired with a residue slot and a following gap slot, allowing the substitution, insertion, and deletion heads to simulate residue-to-residue substitution, gap-to-residue insertion, and residue-to-gap deletion, respectively. Following the route-and-denoise view of Zheng et al. (2023a), we update the current noisy positions, select low-confidence positions for the next noisy set, and renoise them with the chosen noising kernel.

## C.2 Evolutionary sampling with deletion, insertion and substitution

We maintain a noisy index set $\mathcal{N}_t$ during sampling, which tracks the indices of tokens that are considered "noisy" at the current step $t$ and will be updated for the next step $t - 1$. The following steps implement $\tilde{p}_\theta(\mathbf{z}_{t-1}|\mathbf{z}_t, \mathbf{x}_t)$ under the canonical alignment, while operating on the collapsed observed sequence $\mathbf{x}_t$. The denoising process at each iteration step $t$ proceeds through four steps.

**Step 1: Deletion prediction.** We first apply deletion decisions to the current noisy positions. The deletion head is applied to the indices in the noisy set, i.e., $j \in \mathcal{N}_t$. If the model predicts deletion (i.e., $p_\theta^{\mathrm{del}}(\mathbf{x}_t^j) > \tau_{\mathrm{del}}$), the token is removed from the sequence. The noisy set $\mathcal{N}_t$ is updated to reflect the shifted indices of the remaining tokens.

**Step 2: Insertion prediction.** The insertion head scans the current noisy set. If an insertion is predicted at index $j$ (i.e., $p_\theta^{\mathrm{ins}}(\mathbf{x}_t^j) > \tau_{\mathrm{ins}}$), a mask token $\mathbf{m}$ is inserted into the sequence at position $j + 1$. Crucially, since these new tokens lack semantic content, their indices are immediately added to the noisy set $\mathcal{N}_t$.

**Step 3: Substitution prediction, along with the insertion content.** The substitution head makes residue predictions for all tokens, yielding a $\hat{\mathbf{x}}_0$ sampled from $p_\theta^{\mathrm{sub}}(\cdot|\mathbf{x}_t)$ and the corresponding confidence score. We update the indices in the noisy set $\mathcal{N}_t$ of $\mathbf{x}_t$ with $\hat{\mathbf{x}}_0$, including both substituted residues and the contents of newly inserted mask tokens. Then, we update the noisy set for the next step, i.e., $\mathcal{N}_{t-1}$, by selecting the $k_t\%$ tokens with the *lowest* confidence scores, where $k_t\%$ follows a linear decay schedule from $100\%$ to $0\%$.

**Step 4: Renoising.** Finally, we perform renoising for the indices in $\mathcal{N}_{t-1}$. This also prevents the model from collapsing into local optima. For every index $j \in \mathcal{N}_{t-1}$, we sample $\mathbf{x}_{t-1}^{(j)}$ from the noise distribution. This noise distribution can be instantiated as the contextualized evolutionary noising kernel (using the model's own predictions) or the BLOSUM-based kernel, ensuring that the noise state remains consistent with the biologically grounded corruptions used during training.

The full procedure is summarized in Algorithm 1.

## D Related work

**Discrete diffusion model** are diffusion models operating in discrete state space (Sohl-Dickstein et al., 2015; Austin et al., 2021). They noises samples with discrete transition probabilities and learn to denoise them iteratively, in comparison to their continuous counterpart using continuous distribution such as Gaussian kernels (Ho et al., 2020; Song et al., 2020a;b). Various transition kernels have been explored, typically uniform transitions (Austin et al., 2021; Sahoo et al., 2025) and masking (He et al., 2023; Ye et al., 2023a; Zheng et al., 2023a; Sahoo et al., 2024; Shi et al., 2024; Nie et al., 2024). Among

---

**Algorithm 1** Approximate evolutionary sampling with DPLM-Evo

---

**Require:** Trained prediction heads $p_\theta^{\text{del}}$, $p_\theta^{\text{ins}}$, and $p_\theta^{\text{sub}}$, Prior length $L_{\text{init}}$, Steps $T$
 1: **Initialize:** $x_T \sim p_\theta(\cdot \mid \mathbf{m}^{L_{\text{init}}})$, $\mathcal{N}_T \leftarrow \{1, \ldots, L_{\text{init}}\}$
 2: **for** $t = T$ **down to** 1 **do**
 3:    // Step 1: Deletion
 4:    Predict deletion: $\mathcal{D} \leftarrow \{j \in \mathcal{N}_t \mid p_\theta^{\text{del}}(x_t^j) > \tau_{\text{del}}\}$
 5:    $x_t \leftarrow \text{Delete}(x_t, \mathcal{D})$, Update indices in $\mathcal{N}_t$
 6:    // Step 2: Insertion
 7:    Predict insertion: $\mathcal{I} \leftarrow \{j \in \mathcal{N}_t \mid p_\theta^{\text{ins}}(x_t^j) > \tau_{\text{ins}}\}$
 8:    $x_t \leftarrow \text{Insert}(x_t, \mathcal{I}, \text{token} = \mathbf{m})$
 9:    $\mathcal{N}_t \leftarrow \mathcal{N}_t \cup \text{Indices}(\text{InsertedToken})$
10:    // Step 3: Substitution
11:    Sample $\hat{x}_0 \sim p_\theta^{\text{sub}}(\cdot | x_t)$
12:    **for** $j$ in $\mathcal{N}_t$ **do**
13:       $x_t^{(j)} \leftarrow \hat{x}_0^{(j)}$
14:    **end for**
15:    $k_t \leftarrow \text{Schedule}(t)$
16:    $\mathcal{N}_{t-1} \leftarrow \text{TopK\_Lowest\_Confidence}(c, k_t)$
17:    // Step 4: Renoise
18:    For each $j \in \mathcal{N}_{t-1}$, sample $x_{t-1}^{(j)} \sim \pi_{\text{noise}}(\cdot | x_t)$.
19: **end for**
20: **Return:** $x_0$

---

the variants, masked diffusion has attracted the most recent interest for their simplicity (Zheng et al., 2023a; Ou et al., 2024), scalability (Ye et al., 2023a; Nie et al., 2024), and empirical effectiveness, gaining success as protein language models (Zheng et al., 2023b; Wang et al., 2024b;a; Hsieh et al., 2025) and large language models (Ye et al., 2023a; Nie et al., 2025; Ye et al., 2025; Gong et al., 2025).

**Flexible length generative model.** Pre-deciding the length of a answer to many problems can be hard due to the uncertain computation budget and answer shapes required. While autoregressive models, which iteratively produce one token each step, naturally provide flexible length generation, they perform inferiorly in generating data without fixed-order structure such as protein sequences (Zheng et al., 2023b). Non-autoregressive generative models (*e.g.*, diffusion models), on the other hand, does not require presumption on the orders but mostly require preset or predicted answer lengths (Guo et al., 2019; Gu & Kong, 2021). To address this, researchers have explored enabling non-autoregressive models to vary the output lengths by introducing indels operations in their predictions, which are also referred to as edit-based models. As early attempt, Levenshtein Transformer (Gu et al., 2019) studies non-autoregressive machine translations and show edit-based models perform on par with autoregressive models while showing much lower sampling latency thanks to parallel generation. Later progress revisit edit-based models under the formulation of diffusion models (Reid et al., 2022). More recently, DreamOn (Wu et al., 2025) introduces a large diffusion language model with indels as speical tokens in vocabulary to vary the length during sampling, tailored for coding tasks. EditFlow (Havasi et al., 2025), most related to our work, takes a flow-based model perspective to construct indels training signals by aligning the noisy samples, which are interpolations between clean samples and noises, with ground truth target. Although our work also extend fixed-length diffusion to supports indels, we highlight the perspective of diffusion transition kernels and their evolutionary significance. In particular, DPLM-Evo uses a data-dependent contextualized kernel tailored to protein substitutions, while the uniform kernel performs poorly in our ablation (Fig. 3D).

**Protein language model.** Motivated by the success of large language models (LLMs), similar practice has been extended to the development of protein language models. ESM-1b (Rives et al., 2019) utilizes self-supervised masked language modeling on 250 million protein sequences spanning evolutionary diversity, later leading to the development of ESM-2 (Lin et al., 2023) scales further. ProtTrans (Elnaggar et al., 2021), ProteinBERT (Brandes et al., 2022), PRoBERTa (Nambiar et al., 2020), ProtAlbert (Behjati et al., 2022), TAPE (Rao et al., 2019), ProteinLM (Xiao et al., 2021), and CARP (Yang et al., 2022) involve several other representative masked language modeling (MLM) paradigm. These sequence-based PLMs perform competitively with classic methods that rely on multiple sequence alignments, indicating that PLMs have captured some of the evolutionary information from sequences alone. In particular, these protein language models achieve powerful generalization on various downstream tasks involving the secondary and tertiary structures. Recent findings

further showcase their capabilities in predicting protein functions (Meier et al., 2021), structure folding (Lin et al., 2023), and de novo designs (Verkuil et al., 2022). Beyond representation learning, DPLM (Wang et al., 2024b) unlocks the generation capabilities of protein MLM through scalable discrete diffusion training process (Ye et al., 2023b;a; Zheng et al., 2023b), enabling generating high-quality protein sequences. Building upon DPLM, DPLM-2 (Wang et al., 2024a) and DPLM-2.1 (Hsieh et al., 2025) further enhance the model with multimodal understanding and generation capabilities. The series lay the foundation of pretrained models and diffusion algorithms for ours. For guided generation, NOS (Gruver et al., 2023) uses gradient-guided discrete diffusion with a differentiable property predictor for protein optimization, while DPLM-Evo supports any classifier for guidance and the two approaches are complementary. For variant effect prediction, PoET (Truong Jr & Bepler, 2024) trains an ensemble of autoregressive models over MSA families, while EVE (Frazer et al., 2021) uses a variational autoencoder on MSAs. Both rely on MSA inputs, whereas DPLM-Evo operates in a single-sequence setting.

# E    GFP optimization pipeline

This section details the directed-evolution search procedure used for the GFP optimization experiment. Starting from the GFP template sequence, DPLM-Evo iteratively proposes single-site variants, filters invalid candidates, ranks the remaining sequences with the common score terms, and keeps a small beam for the next round. Algorithm 2 summarizes the full loop and the hyperparameters used in our experiments.

---

**Algorithm 2** GFP Directed Evolution by DPLM-Evo

---

1: **Initialization**: Starting from the GFP template sequence, add it to the candidate set $\mathcal{C}$.
2: **Hyperparameters**:
3:     max iteration $T = 20$
4:     search width $w = 100$
5:     beam size $b = 10$
6: **for** $i = 1, \ldots, T$ **do**
7:     **Generate mutated sequences**: For each sequence in $\mathcal{C}$, generate $w$ mutated sequences (one position is mutated at a time).
8:         Obtain a total of $|\mathcal{C}| \times w$ samples.
9:     **Filter candidates**: Filter the generated samples by {Common Filters}.
10:    **Sort candidates**: Sort the filtered candidates according to {Common Score Terms}.
11:    **Update candidate set**: Select the sequences with $\text{top } b$ scores as the next-iteration candidate set $\mathcal{C}$.
12: **end for**
13: **Return**: The final candidate set $\mathcal{C}$

---

# F    Additional Experimental Results

This section collects supplementary quantitative results that complement the main experimental section. For ProteinGym substitution benchmark, we compare DPLM-Evo with baselines that use structure, MSA, or multi-modal information. Meanwhile, we provide assay-level ProteinGym breakdowns to clarify how the aggregate trends vary across benchmarks. For unconditional generation, we report additional generation-quality statistics, including pLDDT comparison with more baselines and secondary structure distribution of the DPLM-Evo generated sequences.

*Table 3. Unconditional generation quality measured by pLDDT.* All baselines except DPLM-Evo are quoted from Meshchaninov et al. (2024).

| Model | pLDDT |
|---|---|
| Dataset (real proteins) | 80.7 |
| DPLM (Wang et al., 2024b) | 84.0 |
| **DPLM-Evo** | **83.6** |
| DiMA (Meshchaninov et al., 2024) | 83.3 |
| nanoGPT | 61.0 |
| RITA | 43.9 |
| SeqDesign | 43.1 |
| DFM | 37.8 |
| EvoDiff-OADM | 37.1 |
| D3PM | 36.7 |
| Walk-Jump | 32.4 |
| proteinGAN | 30.4 |
| Random sequences | 24.8 |

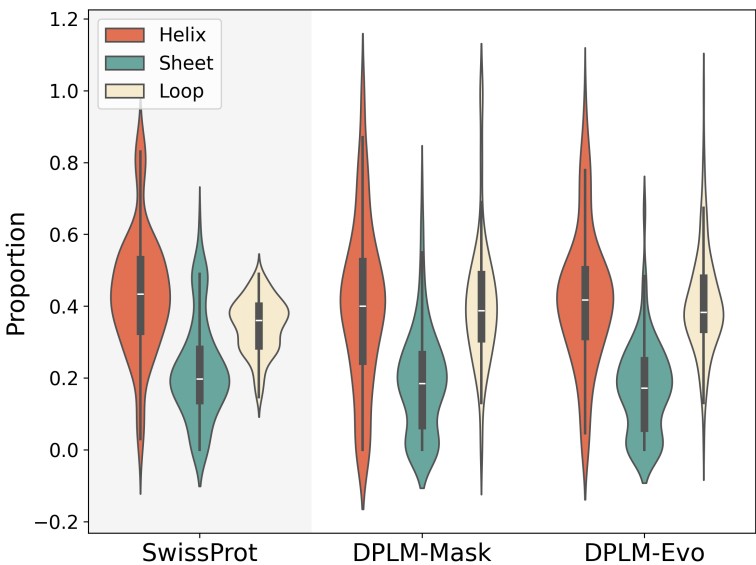

*Figure 7. Secondary structure distribution comparison across SwissProt, DPLM-Mask, and DPLM-Evo.* We computed the proportions of helix, sheet, and loop for each generated protein on ESMFold-predicted structures, alongside an equal-sized random sample from SwissProt. Both DPLM-Mask and DPLM-Evo closely resemble the natural secondary structure distribution of SwissProt. While DPLM-Mask shows a comparable match in secondary structure proportions, DPLM-Evo exhibits superior diversity and reduced mode collapse as shown in Fig. 3.

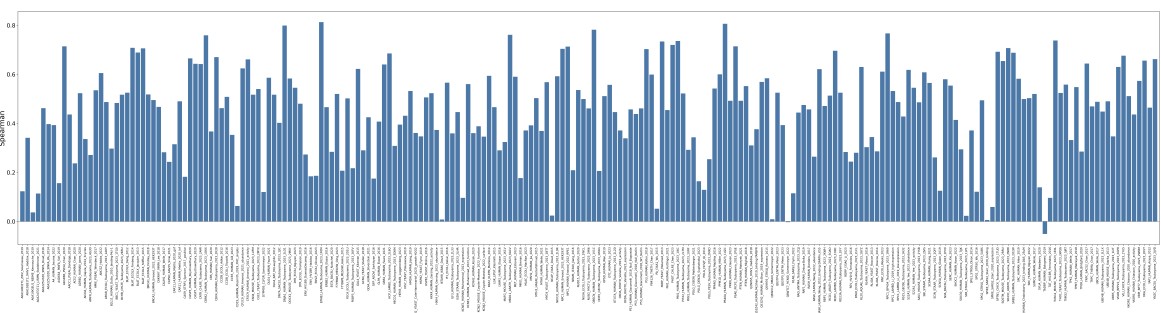

*Figure 8.* Assay-level Spearman correlations of DPLM-Evo across 217 ProteinGym DMS substitution assays; higher values indicate better agreement with experimental measurements.

*Table 2. ProteinGym substitution benchmark: additional baselines using structure, MSA, or multi-modal information.* These methods fall outside the single-sequence setting targeted by DPLM-Evo (0.420 Spearman, 0.459 with GEMME alignment).

| Method | Input Modalities | Avg. Spearman |
|---|---|---|
| AIDO Protein-RAG (16B) | Structure & MSA | 0.518 |
| VenusREM | Structure & MSA | 0.518 |
| ProSST (K=2048) | Seq. & Structure | 0.507 |
| S3F-MSA | Structure & MSA | 0.496 |
| Protriever | MSA | 0.479 |
| ESCOTT | Structure & MSA | 0.476 |
| S3F | Seq. & Structure | 0.470 |
| ProFam (ensemble) | MSA | 0.470 |
| PoET (200M) (Truong Jr & Bepler, 2024) | MSA | 0.470 |
| ESM3 open (1.4B) (Hayes et al., 2024) | Seq. Str. & Func. | 0.466 |
| DPLM-Evo w/ alignment | Seq. | 0.459 |
| GEMME (Laine et al., 2019) | MSA | 0.455 |
| EVE (ensemble) (Frazer et al., 2021) | MSA | 0.439 |
| DPLM-Evo | Seq. | 0.420 |

