# OpenReview forum: "Towards A Generative Protein Evolution Machine with DPLM-Evo"
_ICML.cc/2026/Conference — ICML 2026 regular_

### Official Review · Reviewer_6inr · 2026-02-21

**Soundness:** 4
**Presentation:** 3
**Significance:** 4
**Originality:** 4
**Overall Recommendation:** 6
**Confidence:** 5

**Summary:**

This paper extends the DPLM framework for discrete diffusion-based protein language models, which uses a denoising model (gradually reversing the incremental masking of a target sequence with mask tokens) to generate a biologically plausible protein sequence, to allow noise in the form of insertions and deletions (indels) as well as substitutions (which can include replacement of amino acids with other amino acids, as well as mask characters). In contrast to EditFlow, these are modeled as a discrete-time process rather than by flow matching. The atomic edits for noise introduction are motivated by biological realism, since indels and substitutions are commonly observed mutations in real protein sequences. The new method (DPLM-Evo) performs favorably on the ProteinGym benchmark, including when compared to some methods that accept additional homology information (e.g. TranceptEVE with a supplementary MSA) or structural information (e.g. SaProt). It also appears to generate plausible (foldable) protein sequences, as well as plausible variants of existing protein sequences, in the sense that those variants are predicted (computationally) to fold to similar structures as the original sequence even when sequence identity is <50%. It performs well on a scaffolding task.

**Compliance With Llm Reviewing Policy:**

Affirmed.

**Final Justification:**

Taking into account the authors' rebuttal I find this a very strong paper.

**Key Questions For Authors:**

Can you clarify what are the implications of fixing the alignment length at 2L? Does this fix, or cap, the number of indels that can be introduced?

Can you clearly state how many parameters the various heads have?

**Limitations:**

The authors claim "Our work on protein generation and representation learning can be used in developing potent therapeutic macromolecues [sic] such as antibodies and accelerate the research process of drug discovery." This is admittedly listed as a "potential" consequence, but it seems to ignore that antibody design is a highly specialized problem. And yes, protein design could accelerate drug discovery, but there are a lot of additional steps there and it's not really a proven claim, more just generic hype.

"Our method may be adapted to other scenarios of computer-aided design, such as small molecule design, material design, and chip design." ... this seems like a stretch to me, without any discussion of how.

"It is also needed to ensure the responsible use of our method and refrain from using it for harmful purposes." One could spell out these purposes. For example, tools for generating diverse variants could be used to circumvent biosafety guards.

**Strengths And Weaknesses:**

The submission appears technically sound. Claims are supported by ProteinGym (a common way of evaluating substitution variants) and by structure prediction. A caveat is that the method, despite using indels as noise, is not evaluated as a predictor of indel variants. This may be because the method only models indels as single-residue mutation events. It is well known in the field of computational biology that such models tend to underestimate the frequency of multi-residue indels (since they consider them to be multiple single-residue indel events). So this is also a weakness of the claim to biological plausibility of the noise mechanism: the restriction to single-character indels somewhat undercuts the claims to emulate the molecular evolutionary and bioengineering design processes, which use multi-character indels. Regardless, this is a strong contribution.

Another technical limitation is that the alignment is constrained to be 2L, twice as long as the initial sequence L. It was a bit unclear to me, reading the text, whether this means that the net number of deletions minus the number of insertions has to be L (for example is that implicit in the Gamma function indicating the distribution over alignments, used in the ELBO?). Certainly it precludes situations where the number of deletions is more than half the initial sequence, which might be a limitation (it’s not really explored or justified as far as I can see).

The presentation is reasonably clear. The meaning of “insertion” and “deletion” can be a bit confusing, as it was unclear to me at first if the direction of time is the denoising direction or the noise direction. Perhaps this is more standard in the discrete diffusion literature. The presentation is perhaps overly jargon-loaded in places. For example the novel phrase “contextual evolutionary noising kernel” is invented for what most computational biologists would just call a substitution matrix (or at least an application of one). But this is a stylistic criticism and does not detract from the impressive results.

Another issue with presentation is that I found it very hard to figure out how many parameters the various different parts of the model had. The only parameter counts I can see are in the x-axes of Figure 2, which mention "DPLM-Evo-3B" and "DPLM-Evo-650M", and this doesn't explain how these parameters are distributed across the substitution & indel heads.

Arguably a fundamental limitation is that the work mostly ignores the substantial existing theory in the field of molecular evolution on modeling such processes. The BLOSUM matrix, for example, appeared in 1992 and was not the last word on the matter. Further, BLOSUM is not a time-dependent process. Thus the noise schedule here, which is tied to the BLOSUM matrix combined with single-character indel rates for each discrete timestep, is a little idiosyncratic and arbitrary. However, this is somewhat tangential to their results and I do not see this as a reason to obstruct publication of work that has apparently good results.

---

> ### Author Rebuttal · Authors · 2026-03-31
>
> We thank the reviewer for the thorough and constructive evaluation. We are encouraged that you view the work as a strong contribution, and we address the specific questions about indel evaluation, 2L alignment, parameter counts, and BLOSUM below.
>
> > [W1] Despite using indels as noise, is not evaluated as a predictor of indel variants. This may be because the method only models indels as single-residue mutation events.
>
> Thank you for raising this. DPLM-Evo achieves 0.495 on the ProteinGym-indel benchmark, compared with 0.465 for the strongest single-sequence baseline ProGen2 M. Detailed results are in [t2](https://anonymous.4open.science/r/DPLM_Evo/t2.md).
>
> > [W2] The restriction to single-character indels somewhat undercuts the claims to emulate the molecular evolutionary and bioengineering design processes, which use multi-character indels.
>
> We acknowledge this limitation. Our formulation models indels as binary per-step decisions, so multi-residue indels emerge iteratively across denoising steps. This captures indel events at the modeling level, though it does not reproduce the full statistical structure of multi-character indels observed in nature. We will adjust the wording accordingly in the revision.
>
> > [W3 & Q1] Another technical limitation is that the alignment is constrained to be 2L. Does this fix, or cap, the number of indels that can be introduced?
>
> Thank you for this question. The fixed 2L latent canvas is for deriving the one-step training objective; it does not constrain the sampling procedure. During iterative sampling, edits accumulate across denoising steps without a 2L cap. We will clarify this distinction in the revised manuscript.
>
> > [W4] The presentation is reasonably clear. The meaning of “insertion” and “deletion” can be a bit confusing, as it was unclear to me at first if the direction of time is the denoising direction or the noise direction. Perhaps this is more standard in the discrete diffusion literature.
>
> Thank you for the feedback. In the revision, we will explicitly state whether insertion and deletion refer to the forward (noising) or reverse (denoising) direction, and maintain that convention consistently.
>
> > [W5 & Q2] Another issue with presentation is that I found it very hard to figure out how many parameters the various different parts of the model had.  Explain how these parameters are distributed across the substitution & indel heads.
>
> Thank you for pointing this out. The DPLM-650M has the shared 650M backbone and three heads (substitution, deletion and insertion head) for substitution logits and indel operations. Each head consists of two MLP layers. We will include this breakdown in the revision as well as further architecture ablations.
>
> > [W6] The noise schedule here, which is tied to the BLOSUM matrix combined with single-character indel rates for each discrete timestep, is a little idiosyncratic and arbitrary.
>
> Thank you for raising this. We appreciate the reviewer's perspective from the molecular evolution literature. To clarify, BLOSUM is not the kernel used by DPLM-Evo. It is included only as an alternative noising kernel for comparison. The main model uses the contextual on-policy kernel, which is data-dependent and sequence-context-aware.
> We will clarify this distinction in the revision and report the comparison result: uniform kernel 0.295, BLOSUM kernel 0.35, compared to contextual kernel 0.42. The conclusion drawn from comparison is: uniform < BLOSUM < contextualized. We attribute this to the differences among the three noise kernels. The uniform kernel adds noise randomly, where the noise carries no evolutionary significance, resulting in biologically implausible noisy inputs from which the model can hardly learn meaningful evolutionary relationships. BLOSUM provides statistically grounded mutation probabilities; however, it is data-independent, the noise depends solely on the current amino acid. While it encourages the model to capture mutational correlations between amino acids, it lacks global, sequence-level consideration. The contextualized kernel, on the other hand, produces data-dependent noise that is specific to the current sequence and evolutionarily meaningful. This allows the model to learn how the same amino acid mutates differently in varying proteins, effectively capturing evolutionary and homologous relationships, and ultimately achieving the best performance on the mutational effect prediction task.
>
> > [Limitations] Impact statement wording
>
> Thank you for the candid feedback. We agree the impact statement should be more grounded. Antibody design involves specialized constraints beyond our current scope, and the extension to small molecules, materials, and chips was speculative. We will narrow the impact statement to capabilities directly demonstrated in the paper. For responsible use, we will spell out the dual-use risk that tools generating diverse variants could be used to circumvent biosafety guards.

---

> > ### Author Rebuttal · Reviewer_6inr · 2026-04-06
> >
> > Thanks for the informative rebuttal. I will increase my score.

---

> > > ### Author Response · Authors · 2026-04-07
> > >
> > > Dear Reviewer 6inr,
> > >
> > > Thank you for reading our rebuttal and for your supportive words! We're happy that we have been able to resolve all your comments, which are super inspiring and have indeed helped greatly enhance our paper. We are once again sincerely grateful and encouraged by the recognition, and will ensure all these improvements are clearly reflected in the final version.
> > >
> > > Best,
> > >
> > > Authors

---

### Official Review · Reviewer_C3nZ · 2026-03-01

**Soundness:** 3
**Presentation:** 3
**Significance:** 3
**Originality:** 3
**Overall Recommendation:** 4
**Confidence:** 4

**Summary:**

The authors propose an edit-based discrete diffusion framework to model protein sequences in UniRef50. Similarly to EditFlows [1], they leverage a site-wise interpolant in a latent alignment space to get noisy samples $x_{t}$ and train the model with a Bregman divergence loss that increases the probability of edit operations that bring $x_t$ closer to the clean sample $x_0$. In terms of novel contributions, the authors devise a bootstrapped contextual noising process that uses the substitution probabilities learned by the model itself to replace amino acids in the forward noising process rather than using an uninformative uniform substitution process. The performance of their method, DPLM-Evo, is thoroughly evaluated on variant effect prediction (ProteinGym), unconditional sequence generation, motif scaffolding, and directed evolution tasks.

**Compliance With Llm Reviewing Policy:**

Affirmed.

**Final Justification:**

My concerns are partially addressed. I maintain my score of 4.

**Key Questions For Authors:**

1) it is not clear to me why the marginal distribution at $t=1$ of the contextual forward noising kernel is well-approximated by
$p_{\theta}(\cdot \mid \mathbf{m}^{L})$. Could the authors provide further discussion (or ideally a proof) that this holds?

2) the *predict-then-renoise* sampling strategy does not seem fully theoretically justified. The authors claim to sample from $p_{\theta}(\hat{x_0} \mid x_{t})$, but I don't think this necessarily true since the denoiser does not predict the probability of multiple consecutive insertion or deletion events at the same site. For instance, in the example $[A, B, C] \to [A, \phi, \phi, B, \phi, C]$, how is $\hat{x_0}$ in the support of $p_{\theta}(\cdot\mid x_{t})$ if $x_{t}=\Gamma^{-1}([A,\phi,\phi,\phi,\phi,\phi])=[A]$?

3) in a similar vein, how does one choose $\tau_{\text{del}}$ and $\tau_{\text{ins}}$ such that the $\hat{x_0}$ is sampled from $p_{\theta}(\cdot\mid x_{t})$ as claimed?

4) Why not report DMS performance on indels in ProteinGym? A major advantage of the new model seems to stem from its ability to model indels.

**Limitations:**

- no limitations of the method are discussed in the main text
- for instance, the additional computational complexity of the on policy contextual noising should be discussed
- i am open to increasing my score if the key questions above are adequately addressed

**Strengths And Weaknesses:**

## Strengths
- The paper is clearly presented and well-structured
- The paper tackles an important limitation of fixed-length discrete diffusion protein sequence generative models
- The on-policy contextual noising is novel (to my knowledge) and well motivated if we assume a time-reversible evolutionary process
- In-silico experimental validation of the model is extensive, with many protein design tasks explored

## Weaknesses
- I have a few concerns with potential mismatches between the presented theory and the practical implementations of the model (See Questions)
- For the directed evolution of GFP, it would be nice to compare against mutations proposed by a stronger pLM baseline like ESM2 rather than randomly sampling mutations
-  The setup of using an interpolant to define the transition kernel and a data dependent noise distribution is more akin to flow matching than discrete diffusion. The current write-up is a little confusing when written in this diffusion perspective.
- Novel algorithmic contributions are somewhat limited since the core model is largely an application of the EditFlows formulation.

## Suggestions
- Results in [1] should be discussed. they train a discrete diffusion model with BLOSUM noising and investigate the performance gap between custom noising kernels and the masked noising kernel, which could explain why the uniform masking strategy performs worse in Fig. 3d
- Can the authors ablate the contextual evolutionary kernel on DMS performance? I am curious where the gain is coming from compared to DPLM-Mask since the eval is restricted to substitutions.
- Some experimental details are missing such as the timestep used for DMS scoring and directed evolution results

---

> ### Author Rebuttal · Authors · 2026-03-31
>
> We thank the reviewer for the careful technical reading. The concerns around theoretical precision, EditFlows overlap, and missing evaluations are addressed below.
>
> > [W1] GFP directed evolution ESM2 baseline
>
> Please kindly refer to Reviewer RQkX Q4 due to length limit.
>
> > [W2] Flow perspective
>
> We thank the reviewer for providing the flow matching perspective to interpret DPLM-evo. Both diffusion (RDM/MDLM/UDLM) and flow matching can be defined in an interpolation form. DPLM-evo is framed as diffusion because we are inspired by mutations in biological sense. DPLM-evo also uses different training objective (ELBO vs. Bregman divergence) and a contextual on-policy kernel instead of a uniform one. We will clarify the connections in the revision with genuine distinctions beyond presentation.
>
> > [W3] Difference from EditFlows
>
> We appreciate this observation. While DPLM-Evo and EditFlows both support edit operations, they are independent works built on different foundations: discrete-time diffusion with ELBO (ours) and contextual kernel versus continuous-time flow matching with Bregman divergence (EditFlows) and uniform source dist. (aka. uniform kernel).
> EditFlows targets text and code, and has not been demonstrated on biological sequences like proteins, which is non-trivial and presents distinct challenges: a constrained fitness landscape and strong non-causal contextual dependencies. Our ablation (Fig. 3d) shows that data independent uniform kernel cause training collapse on proteins. The contextual on-policy kernel resolves this with data-dependent, sequence-context-aware corruption (Spearman 0.295 → 0.42), and has no counterpart in EditFlows. The discrete-diffusion formulation further enables finetuning from a pretrained masked DPLM.
>
> > [Q1] contextual kernel marginal distribution
>
>
> Thank you and this is a fair point that we should have made clearer.
>
> We start with clarifying the definition of the $Q\_\text{noise}$ in our submission.
> Consider a length-$L$ protein $x\_{0}$, its corresponding latent seuquence is $z\_{0}$, a specific amino acid $v$ at position $j$, and let ${z'}\_t^{\setminus j}$ denote the auxiliary sequence with position $j$ removed.
> Specifically, we redefine the $U\_{K}$ in the top-left block of $Q\_\text{noise}$ as a more generalized substitution matrix $T\_\text{sub}$, yielding different design choices $T\_\text{sub}$:
> - when $T=U\_{K}$, it gives uniform kernel;
> - when $T=M\_{\text{BLOSUM}}$ in Eq.5, it gives the (context-independent) BLOSUM kernel; and
> - when $T=\mathbb{E}\_{q'\_{t}(z'\_{t}|z\_{0})}[p\_{\theta}(v|{z'}\_{t}^{\setminus j}, \mathbf{m})]$, where $q'^{(j)}\_{t}(\cdot|z\_{0}) = \bar{\alpha}\_{t} \delta\_{z\_{0}^{(j)}} + (1{-}\bar{\alpha}\_{t})\mathbf{m}$ is a masked-diffusion auxiliary process, it leads to the data-dependent contextualized kernel.
>
> As such, the resulting forward process is then $q^{(j)}\_{t}(v|z\_{0}) = \bar{\alpha}\_{t} \delta\_{z\_{0}^{(j)}}(v) + (1{-}\bar{\alpha}\_{t})\pi(z\_0)$, where $m^{j}$ denotes a single mask at position $j$.
> By definition of $q\_t(\cdot|z\_{0})$, at $t{=}1$ (fully noised), $\bar{\alpha}\_{1}{=}0$ and $z'\_{1} \sim q'\_{t=1}(\cdot|z\_{0}) = \mathbf{m}^{L}$ deterministically. Therefore, $q\_{1}^{(j)} = p\_{\theta}(v|\mathbf{m}^{L})$.
>
> *Practical intuition*: following MDLM/DPLM etc, generation starts from all-mask and uses $p\_{\theta}(\cdot|m^L)$ as a learned prior that captures unconditional AA frequencies.
>
> We will add more rigorous discussion and clarification in the revised version. Thank you so much!
>
> > [Q2] Predict-then-renoise sampling justification
>
> Thank you for this concrete example. The predict-then-renoise strategy follows the RDM route-and-denoise decomposition (Zheng et al., 2023; we refer the reviewer there for details). In training, the model learns to reverse only one denoising step by design. Since $\mathbf{Q}_{\text{noise}}$ permits at most one operation per position per step, including at most one insertion, per-position predictions are sufficient for single-step reverse transitions. In generation, multi-residue insertions are naturally decomposed into iterative decoding across steps for simplicity. The reviewer's [A, B, C] example is exactly such a multi-residue indel that accumulates across denoising steps, consistent with this one-edit-per-step design.
>
> > [Q3] Indel sampling hyperparameter
>
> Following from [Q2], these thresholds are practical inference hyperparameters. We set $\tau\_{\text{ins}} = \tau\_{\text{del}} = 0.7$ to favor conservative edit decisions during generation, avoiding excessive insertions or deletions per step.
>
> > [Q4] ProteinGym indel benchmark
>
> Thank you for raising this. We agree this is essential given our indel-modeling contribution. We conducted exp. as suggested, where DPLM-Evo obtains 0.495, vs 0.465 the currently strongest model ProGen2 M.
>
> On Suggestions, thank you and please kindly refer to the following responses:
> * S1: Reviewer 6inr W6
> * S2: Q4
> * S3: Reviewer C3nZ Q4

---

> > ### Author Rebuttal · Reviewer_C3nZ · 2026-03-31
> >
> > I thank the authors for their rebuttal. I have a few follow-up questions:
> >
> > 1. In the directed evolution of GFP experiment, do you sample indels using DPLM-Evo or only perform substitutions? The performance of ESM2 is quite competitive (0.737 vs 0.793 pTM) and i wonder if DPLM-Evo can do better with indels.
> >
> > 2. I am not convinced by the *predict-then-renoise* sampling justification. Sampling from Eq. 8 is intractable with indels, so it seems like the authors are performing some kind of approximate sampling based on heuristic choices for target insertion and deletion rates $\tau_\text{ins,del}$. The Zheng. et al paper requires us to draw $\hat{x_0}$ which is doable for substitution only process, but not for indel process. Could the authors clarify this?
> >
> > 3. Similarly, I believe that the provided justification for the prior distribution under the contextual kernel checks out for substitution only process, but it is unclear to me that the proof holds under non-zero indel rates in the forward process. Can the authors provide clarification here?

---

> > > ### Author Response · Authors · 2026-04-06
> > >
> > > > [Q1]
> > >
> > > Thank you for the question.
> > > The original experiment used sub-only sampling to match ESM-2 settings.
> > > With DPLM-Evo's indel sampling enabled, the final pTM reaches 0.857, a clear improvement over both ESM-2 and the sub-only variant.
> > > Moreover, DPLM-Evo with indels surpasses ESM-2's final pTM at step 7 out of 20 (65% faster), while the sub-only variant reaches this level at step 13 (35% faster).
> > > We include the updated comparison figure in the revision [fig4](https://anonymous.4open.science/r/DPLM_Evo/f4.pdf).
> > >
> > > > [Q2]
> > >
> > > Thanks for the insightful question. We clarify this below from both a theoretical and a practical perspective.
> > >
> > > **Theoretical perspective**: In the proposed evolutionary discrete diffusion framework, the reverse marginal in Eq. 8 can be extended to the latent alignment space as in [e1](https://anonymous.4open.science/r/DPLM_Evo/e1.pdf),
> > > where $x_{t-1}$ is deterministically obtained from $z_{t-1}$ via $\Gamma^{-1}(z_{t-1})$, i.e., by removing all $\phi$ tokens in $z_{t-1}$.
> > > The variables $z_t$ and $z_{t-1}$ are summed over $\Gamma(x)$, which is the set of latent alignments that collapse to $x$.
> > > Here the gap token $\phi$ is an additional latent state in $\mathbf{Q}_{\text{noise}}$, so $\phi \to \mathrm{AA}$ and $\mathrm{AA} \to \phi$ naturally represent indels.
> > > Hence Eq. 8 admits an analogous route-and-denoise factorization proposed in Zheng et al. as standard discrete diffusion in latent space.
> > > However, exact sampling from Eq. 8 is still intractable because it requires marginalizing over all latent alignments in $\Gamma(x_t)$.
> > >
> > > **Practical considerations**:
> > > Accordingly, Algorithm 1 is a practical approximate sampler for Eq. 8 rather than an exact sampler, combining both a deterministic canonical alignment and binary classification for indels. Concretely, we make the following implementation choices:
> > > - Canonical alignment: we fix latent alignment to a single canonical form with one insertion slot per residue, eliminating the sum over $\Gamma(\mathbf{x}_t)$. and the $p(\mathbf{z}_t|\mathbf{x}_t)$ is a deterministic distribution:
> > >   - If $\mathbf{x}_t=[\text{A},\text{B},\text{C}]$, then $\mathbf{z}_t=[\text{A}, \phi,\text{B}, \phi,\text{C}, \phi]$
> > > - Single-step indels: due to the canonical alignment, each denoising step allows at most one insertion per position, with multi-residue insertions are approximated by composing repeated edit decisions accumulating across steps (empirically supported by Fig. 3B–C).
> > >   - Therefore, for the reviewer’s example, the sampler is not claiming to draw $[A, \phi, \phi, B, \phi, C]$ from one step conditioned on $x_t = [A]$. Rather, such multi-residue insertions are performed progressively through repeated insertions over successive reverse steps.
> > > - Binary classification as workaround for indels: We formulate indels as binary prediction due to the potential mode collapse issue as described in Appendix A.2. Thus, $\tau_{\text{ins}}$ and $\tau_{\text{del}}$ are not target indel rates in the diffusion kernel; they are inference-time thresholds used to discretize the predicted edit probabilities.
> > >
> > > We will clarify this distinction in the revision. Thank you so much!
> > >
> > > > [Q3]
> > >
> > > Thank you for pointing this out. We agree that the argument in Appendix A.1.5 only justifies the substitution part of the fully noised distribution. It does not imply when the indel rates are non-zero, the prior is exactly equal to $p_{\theta}(\cdot \mid \mathbf{m}^{L})$.
> > > On the other hand, we show that when the indel rates are chosen symmetrically, i.e. $\omega_{\text{ins}}=\omega_{\text{del}}$ (that is what we did during training), the expected observed length is equal to L. Starting from a length $L$ sequence, the canonical latent alignment has $L$ AA and $L$ gaps. For an AA, the probability of remaining non-gap at step $t$ is $1-(1-\bar{\alpha}\_t)\omega\_{\text{del}}$, while for a gap, the probability of becoming non-gap is $(1-\bar{\alpha}\_t)\omega\_{\text{ins}}$. Therefore,
> > > $$\mathbb{E}[|x_t|]=L\big(1-(1-\bar{\alpha}\_t)\omega\_{\text{del}}\big)+L(1-\bar{\alpha}\_t)\omega\_{\text{ins}}=L[1+(1-\bar{\alpha}\_t)(\omega\_{\text{ins}}-\omega\_{\text{del}})].$$
> > > Hence, if $\omega\_{\text{ins}}=\omega\_{\text{del}}$, then $\mathbb{E}[|x\_t|]=L$ for all $t$, and in particular $\mathbb{E}[|x\_T|]=L$. This only justifies the expected length, not the full prior distribution. However, this motivates our practical use of $p_{\theta}(\cdot\mid\mathbf{m}^L)$ as the prior, with the remaining length variation handled by the insertion/deletion heads during reverse denoising.
> > >
> > > We will carefully revise the manuscript to make this distinction explicit and to avoid any misleadings that the prior itself captures the length distribution under non-zero indel rates.
> > >
> > > ---
> > > We would like to sincerely thank the reviewer for the constructive feedback that significantly improved the quality of this paper. We will ensure all these improvements are clearly reflected in the final version. MANY THANKS!
> > >
> > > Best,
> > >
> > > Authors

---

### Official Review · Reviewer_RQkX · 2026-03-04

**Soundness:** 2
**Presentation:** 3
**Significance:** 3
**Originality:** 3
**Overall Recommendation:** 5
**Confidence:** 3

**Summary:**

The authors propose DPLM-evo, a discrete diffusion-based framework that models substitution, insertion, and deletion on protein sequences. It also introduces a so-called contextual evolutionary noising kernel for the forward diffusion process. Benchmarked on ProteinGym and an unconditional sequence generation benchmark, DPLM-evo is claimed to achieve state-of-the-art (SOTA) performance on the ProteinGym benchmark.

**Compliance With Llm Reviewing Policy:**

Affirmed.

**Final Justification:**

The author has addressed my concerns.

**Key Questions For Authors:**

1. According to the authors, “DPLM-Evo achieves state-of-the-art mutation effect prediction on ProteinGym in the single-sequence setting”. Based on the proteinGym benchmark leaderboard [1], there are a few of competitive methods  (such as ProSST, S2/3F, ESM3, PoET, to name a few) not being included in the Figure 2 (the benchmark performance figure), which have better avg spearman score than the proposed DPLM-evo. Could the author explain why these methods are not included? This can weaken the claim of “SOTA”. Also, can you report break-down spearman scores as in ProteinGym as well as the error bar (i.e. std)?
  2. Since one key contribution is the capability of modeling indel. Could the author also benchmark the proposed DPLM-evo on the ProteinGym-indel benchmark, which is highly relevant? When reporting this result, the error bar is expected.
  3. The DPLM-evo seems to use quite larger parameters (650M, 3B) and more compute (training) than other baselines in the ProteinGym, for example, the ProSST only have 110M parameters while achieving better performance than DPLM-evo. Could the authors explain other potential benefits using such a “giant” model for this task?
  4. The case study in section 4.5 is interesting. Could the author elaborate how specifically the Directed Evolution is conducted in this paper? What is Common Filters and Common Score Terms in Algo. 2 ? Can you list a few (irreplaceable) advantages of DE using DPLM-Evo, compared to traditional scoring methods?
  5. Is there substantial advantages of using edit-based diffusion compared to masked discrete diffusion regarding protein sequence generation?

[1] Notin, Pascal, Aaron Kollasch, Daniel Ritter, Lood Van Niekerk, Steffanie Paul, Han Spinner, Nathan Rollins et al. "Proteingym: Large-scale benchmarks for protein fitness prediction and design." Advances in neural information processing systems 36 (2023): 64331-64379.

**Limitations:**

Limitation is not explicitly discussed but encouraged to discuss in terms of (1) model pitfalls, (2) unideal performance, etc.

**Strengths And Weaknesses:**

Pros:
  1. The authors innovatively apply an edit-based diffusion process to protein sequence generation and mutation effect prediction.
  2. The use of standard benchmarks makes the results significant. The paper also includes an interesting case study.

Cons:
  1. The paper is missing related work around edit-based discrete diffusion or flow matching (such as Edit Flows [1]), as well as variant effect prediction (using representative baselines benchmarked in Proteingym [2]).

2. There is insufficient investigation (e.g., an ablation study) demonstrating the necessity of the highlighted contextual evolutionary noising kernel over a uniform diffusion kernel.Incomplete evaluation: There are missing comparisons and evaluations regarding ProteinGym (please see the questions below).


[1] Havasi, Marton, Brian Karrer, Itai Gat, and Ricky TQ Chen. "Edit Flows: Flow Matching with Edit Operations." arXiv preprint arXiv:2506.09018 (2025).

[2] Notin, Pascal, Aaron Kollasch, Daniel Ritter, Lood Van Niekerk, Steffanie Paul, Han Spinner, Nathan Rollins et al. "Proteingym: Large-scale benchmarks for protein fitness prediction and design." Advances in neural information processing systems 36 (2023): 64331-64379.

---

> ### Author Rebuttal · Authors · 2026-03-31
>
> We thank the reviewer for the positive assessment. The main gaps you identified (comparison scope, kernel ablation, indel evaluation) are addressed below.
>
> > [W1] The paper is missing related work around edit-based discrete diffusion or flow matching (such as Edit Flows [1]), as well as variant effect prediction (using representative baselines benchmarked in Proteingym [2]).
>
> Thank you for pointing these out. We cited EditFlow in line 198 and had extended related work section including edit-based literature in appendix (line 793) and will move them to the main manuscript in the revised manuscript. For VEP baselines, we mainly targeted single sequence setting; and will cite others related work in the next revision. Please see our detailed discussion in Reviewer 5XmA W1.
>
> > [W2] There is insufficient investigation (e.g., an ablation study) demonstrating the necessity of the highlighted contextual evolutionary noising kernel over a uniform diffusion kernel.
>
> Thank you for the suggestion. We provided ablations on both generation and scoring. For generation, Fig. 3d has showed that the uniform kernel fails to produce structurally plausible sequences. For ProteinGym scoring, the contextual kernel improves average Spearman from 0.295 to 0.42 over uniform corruption, confirming its contribution across tasks. Please kindly refer to Reviewer 6inr W6 response for additional discussion due to length limit.
>
> > [Q1] Could the author explain why these methods (ProSST, S2/3F, ESM3, PoET) are not included? This can weaken the claim of “SOTA”. Also, can you report break-down spearman scores as in ProteinGym as well as the error bar (i.e. std)?
>
> Thank you for flagging this. Among the mentioned methods: ProSST and S2/3F leverage structure information, ESM3 is a multimodal model, and PoET requires homologous sequences. These fall outside the single-sequence setting and are now reported in separate categories (please kindly refer to Reviewer 5XmA W1 response for additional comparison). We report overall std 0.19 and breakdowns in [fig gym](https://anonymous.4open.science/r/DPLM_Evo/f1.pdf).
>
> > [Q2] Could the author also benchmark the proposed DPLM-evo on the ProteinGym-indel benchmark
>
> Thank you for raising this. We agree this evaluation is essential given that indel modeling is a core contribution. We conducted experiments as suggested, where DPLM-Evo achieves 0.495, compared with 0.465 for the strongest single-sequence baseline ProGen2 M. Benchmark results for other top methods shown in [t2](https://anonymous.4open.science/r/DPLM_Evo/t2.md).
>
> > [Q3] Could the authors explain other potential benefits using such a “giant” model for this task? 650M, 3B compared with ProSST (110M)
>
> Thank you for the question. ProSST benefits from structure information, which is not always available or requires an additional model to predict. A fairer comparison is among single-sequence foundation models. Compared with ESM-2, DPLM-Evo exhibits consistent scalablity for VEP, from 650M to 3B (0.420->0.424 without alignment and 0.450->0.459 with alignment), in ProteinGym benchmark, while ESM-2 regresses at 3B (0.414->0.406, ~0.01 Spearman drop). Additionally, DPLM-Evo natively supports generation, post-editing, and directed evolution beyond substitution scoring.
>
> > [Q4] The case study in section 4.5 is interesting. Could the author elaborate how specifically the Directed Evolution is conducted in this paper? What is Common Filters and Common Score Terms in Algo. 2 ? Can you list a few (irreplaceable) advantages of DE using DPLM-Evo, compared to traditional scoring methods?
>
> The directed evolution follows a beam-search procedure (Algo. 2): starting from the GFP template, each iteration generates 100 mutated sequences per candidate (one position mutated at a time), filters by chromophore site RMSD < 1.5Å (following ESM3), scores by pTM from Chai-1, and retains the top-10 candidates. After 20 iterations, pTM increases from 0.263 to 0.793 while RMSD stays below 1.5Å. As a comparison, an ESM2 baseline can only reach 0.737, and 0.584 for random mutation.
>
> The main advantage of DPLM-Evo: it adaptively proposes both mutation positions and residue identities, rather than relying on fixed heuristics or external rescoring. Compared with ESM-2, it requires an artificial mask-forward-unmask cycle and external position selection heuristics. We additionally provide the whole trajectory of GFP directed evolution pTM comparison in [fig GFP](https://anonymous.4open.science/r/DPLM_Evo/f1.pdf)
>
> > [Q5] Is there substantial advantages of using edit-based diffusion compared to masked discrete diffusion regarding protein sequence generation?
>
> Yes. Substitutions enable natural post-editing without remasking. Indels enable variable-length generation. Neither is natively supported by masking-only diffusion. Our framework interpolates between substitution denoising and mask prediction, so infilling capabilities are retained.

---

> > ### Author Rebuttal · Reviewer_RQkX · 2026-04-04
> >
> > Thank the authors for their response. I will increase my score accordingly.

---

> > > ### Author Response · Authors · 2026-04-04
> > >
> > > Dear Reviewer RQkX,
> > >
> > > Thank you for your time and for confirming that our response have fully addressed your concerns. We sincerely appreciate your continued support and positive feedback.
> > >
> > > Your previous feedback regarding the indel benchmark, taxonomy, and ablation studies was invaluable. Addressing these points clarified our work, and strengthened the paper. We will ensure all these improvements are clearly reflected in the final version.
> > >
> > > Best,
> > >
> > > Authors

---

### Official Review · Reviewer_5XmA · 2026-03-12

**Soundness:** 3
**Presentation:** 4
**Significance:** 3
**Originality:** 3
**Overall Recommendation:** 5
**Confidence:** 3

**Summary:**

This work presents DPLM-Evo, a diffusion PLM designed to model substitutions and indels allowing for variable length generation using latent alignment. DPLM-Evo opts for an alternative noising scheme, parting from standard masked absorbing state diffusion models in favor of evolutionary-based noise. Additionally, DPLM-Evo introduces a variety of techniques to improve training and inference, allowing the model to increase performance over current state-of-the-art methods on a range of relevant tasks.

**Compliance With Llm Reviewing Policy:**

Affirmed.

**Final Justification:**

My final recommendation is to accept (5). In my original review, I was concerned with missing details and required clarification on baseline evaluations. While I would defer to the technical expertise of the other reviewers in regard to any algorithmic concerns, my concerns were addressed and I now have a better understanding of the limitations of the method and where it sits amongst the existing literature. I believe the novelty of the presented methods along with the positive results warrants an accept, and I trust the authors to make necessary edits to improve the paper for a camera-ready publication.

**Key Questions For Authors:**

Can the authors elaborate on the alignment process of DPLM-Evo and GEMME?

What is the motivation for calling the noising method a “contextual evolutionary noising kernel” based on its description? This seems more akin to uncertainty based approaches used in active learning. The proposed BLOSUM-informed substitution seems more aptly suited for this name. On that same note, to what extent did the authors assess this method for the noising procedure?

I was a bit confused on the distinction that this work posed between masked diffusion mechanisms and DPLM-Evo. Is DPLM-Evo not also trained using masked diffusion?

Under what regimes does DPLM-Evo perform best? While average Spearman is insightful to overall performance, I am curious if the authors have insight into when you would choose DPLM-Evo over other models depending on the characteristics of the target protein.

What is the proportion of secondary structures generated (alpha helix %/beta sheet %) during the unconditional generation task?

How does training duration and convergence compare to other diffusion based methods?

**Limitations:**

Discussion on limitations is lacking.

**Strengths And Weaknesses:**

Strengths:

There are clear and valuable contributions of DPLM-Evo. Variable length generation and explicit indel modeling is a valuable technical contribution to the community, neatly resolving a limiting constraint of discrete diffusion models. Uncertainty based masking is an interesting and well-motivated contribution to discrete diffusion corruption. This work assesses a variety of relevant tasks ranging from variant effect prediction to unconditional generation, demonstrating the versatility of DPLM-Evo.

Weaknesses:

Results are benchmarked across a variety of models, but seemingly exclude many relevant models fitting the description of Figure 2. This includes but is not limited to: PoET, Protriever, GEMME, and EVE. Additionally, other benchmarks lack rigorous evaluation, for example in the unconditional generation task.

There is a lack of discussion on limitations and tradeoffs for DPLM-Evo. It is difficult to get a sense of why one might choose DPLM-Evo compared to other models for similar tasks. Along the same lines, descriptions on datasets used and experimental procedures are lacking.

This work is missing a key reference to a seminal work in this area: “Protein Design with Guided Discrete Diffusion” (Gruver et al., 2023).

While it is clear that modeling indels more closely resembles evolutionary processes, at many points this relationship seems overstated. In particular, contribution point 3 claims a biologically-informed noising scheme to improve “learning efficiency and evolutionary consistency” but it is not clear how this work substantiates those claims.

My recommendation is to reject for these reasons. I feel that this work presents many novel and impactful ideas, but does so in a way that overstates the significance or capabilities of DPLM-Evo without proper contextualization to other relevant methods or without proper justification. I am willing to favorably change my score should the authors make a reasonable attempt at addressing my concerns.

---

> ### Author Rebuttal · Authors · 2026-03-31
>
> We thank the reviewer for the thoughtful feedback and for the willingness to reconsider the score. We address each point below.
>
> > [W1] This (ProteinGym) includes but is not limited to: PoET, Protriever, GEMME, and EVE
>
> The SOTA claim targets the single-sequence FM setting (as stated in abstract and §4.1 Line 324). PoET, Protriever, GEMME, and EVE all leverage homologous/evolutionary information, so we now report them in a separate category in [T1](https://anonymous.4open.science/r/DPLM_Evo/t1.md). Within the single-sequence FM setting, DPLM-Evo achieves 0.42 in substitutions (0.459 for 3B model after alignment) with breakdowns in [fig gym](https://anonymous.4open.science/r/DPLM_Evo/f2.pdf). We additionally report single-sequence SOTA in ProteinGym indels benchmark with spearman 0.495, surpassing the second ProGen2 M (0.465) by a large margin. Additional indel baselines shown in [T2](https://anonymous.4open.science/r/DPLM_Evo/t2.md)
>
>
> > [W2]&[Q5] Additional evaluation.
>
> We now report secondary-structure composition in [fig SS](https://anonymous.4open.science/r/DPLM_Evo/f3.pdf), additional generation pLDDT baselines in [T3](https://anonymous.4open.science/r/DPLM_Evo/t3.md).
>
> > [W3]&[Q4] Discussion on DPLM-evo limitation & tradeoffs
>
> DPLM-Evo is most advantageous for post-editing, variable-length generation, and joint substitution-indel tasks, where explicit edit operations offer capabilities that masking-only diffusion does not support natively. Tradeoffs include contextual noising overhead during training and multi-stage training.
>
> > [W4] Descriptions on datasets used and experimental procedures are lacking.
>
> The DPLM-Evo is trained on the UniRef50 dataset. For ProteinGym substitution benchmark, we leverage $\log p(\mathbf{x}\_t=\text{mut}|\mathbf{x}) - \log p(\mathbf{x}\_t=\text{wt}|\mathbf{x})$ for scoring. For indels, we first compute the Levenshtein operations between wt and mut and also calculate the probability of performing an indel operation minus the probability of not performing: $\log p(\text{del}) − \log p(\text{keep}) = \log \text{sigmoid}(l) −
> \log \text{sigmoid}(−l) = l$, where $l$ is the logit in the case of delete. An analogous formulation is used for insertions. For generation, we leverage a predict-then-renoise strategy (details in Appendix B). The threshold of the deletion and insertion head is set to 0.7 for more stable generation.
>
> > [W5] Missing NOS reference
>
> Thank you for pointing this out. NOS (Gruver et al., 2023) uses gradient-guided sampling, requiring a differentiable property predictor trained specifically for this model, while DPLM-Evo supports any kind of classifier for guidance in the edit-based denoising process. The two approaches are complementary and NOS can be integrated to DPLM-evo.
>
> > [W6] Overstated evolutionary resemblance
>
> Our argument is that explicit edit operations (substitutions, insertions, deletions), together with context-dependent corruption, better match protein engineering workflows than masking-only corruption. "Learning efficiency" is supported by ProteinGym Spearman (uniform kernel 0.295 → contextual kernel 0.42), and "evolutionary consistency" by family expansion producing structurally preserved (RMSD < 4) yet diverse (identity < 50%) sequences.
>
> > [Q1] Elaborate on the alignment process of DPLM-Evo and GEMME
>
> The alignment here is kernel-level: following VespaG, we minimize $\mathcal{L}\_{\text{align}} = \sum\_t D\_{\mathrm{KL}}(p\_\theta(\cdot | \mathbf{x}, t) \| p\_{\text{GEMME}}(\cdot | \mathbf{x}))$ over masked positions at each timestep to finetune DPLM-Evo substitution distribution toward the GEMME-derived evolutionary distribution. This further improves mutation effect prediction.
>
> > [Q2] Contextual kernel difference from BLOSUM and ablation
>
> Please kindly refer to Reviewer 6inr W6 response due to length limit.
>
> > [Q3] DPLM-evo difference from mask diffusion
>
> Our framework generalizes existing discrete diffusion through $\mathbf{Q}\_{\text{noise}}$. We have related descriptions in line  176, and we provide more details here. Setting $\omega\_{\text{del}}=0, \omega\_{\text{ins}}=0$ disables indels and recovers a fixed-length model; further setting $\rho\_{\text{mask}}=1$ recovers classical masked diffusion, $\rho\_{\text{mask}}=0$ recovers uniform diffusion, and $\rho\_{\text{mask}} \in (0,1)$ gives a mixed masked-uniform path. DPLM-Evo operates with all coefficients active, adding explicit substitution, insertion, and deletion while retaining mask prediction capabilities.
>
> > [Q6] How does training duration and convergence compare to other diffusion based methods?
>
> Contextual kernel takes 0.875s per step vs 0.705s for uniform (+24.26% overhead), this is because we leverage model prediction to construct $x_t$, so there is an additional forward process (this process does not need gradients). For the total training cost, we use 32 H100 GPUs and always run 100K training steps, which takes about 25 hours.

---

> > ### Author Rebuttal · Reviewer_5XmA · 2026-04-02
> >
> > I appreciate the thoroughness of the responses. I will increase my score.

---

> > > ### Author Response · Authors · 2026-04-03
> > >
> > > Dear Reviewer 5XmA,
> > >
> > > Thank you for reading our rebuttal. We are very glad that our responses have addressed all of your concerns, and we sincerely appreciate your encouraging feedback and the increased rating.
> > >
> > > We would like to once again thank you for your insightful comments, and we will certainly make every effort to incorporate these clarifications, discussions, and new results more clearly in the final version.
> > >
> > > Best,
> > >
> > > Authors

---

### Decision · Program_Chairs · 2026-04-30

**Decision:**

Accept (regular)

**Comment:**

The reviewers agree that this paper presents a technically sound and original extension of discrete diffusion protein language models, introducing explicit modeling of substitutions and indels with strong empirical performance across multiple protein modeling tasks. In particular, the reported results on ProteinGym, including the indel benchmark, and the demonstrated capabilities for generation, post‑editing, and directed evolution constitute clear strengths.

Several reviewers initially raised concerns regarding theoretical justification (especially for sampling with indels), framing relative to prior work in molecular evolution and edit‑based diffusion, and completeness of evaluation. After carefully reviewing the rebuttal and discussion, I find that these concerns have been substantially addressed. The authors provided additional experiments, clarifications of approximations, ablations supporting the contextual noising kernel, and a more transparent discussion of limitations. While some theoretical aspects remain approximate, this is now clearly acknowledged and does not undermine the overall soundness of the work.

I have read and taken into account the authors’ rebuttal and the reviewers’ follow‑up comments in reaching this decision. Overall, the paper is well executed, non‑redundant with prior work, and likely to be useful to a meaningful segment of the ICML community. I therefore recommend acceptance.